# Bedrock ledges, colluvial wedges, and ridgetop wetlands: Characterizing geomorphic and atmospheric controls on the 2023 Wrangell landslide to inform landslide assessment in Southeast Alaska, USA

Joshua J. Roering*[1], Margaret M. Darrow[2], Annette I. Patton[3], Aaron Jacobs[4]

[1] Department of Earth Sciences, University of Oregon, Eugene, OR, USA
[2] Department of Civil, Geological, and Environmental Engineering, University of Alaska Fairbanks, Fairbanks, AK, USA
[3] College of Forestry, Oregon State University, Corvallis, OR, USA
[4] National Weather Service, Juneau, AK, USA

*Correspondence to*: Joshua J. Roering (jroering@uoregon.edu)

**Abstract.** In the past decade, several fatal landslides have impacted Southeast Alaska, highlighting the need to advance our understanding of regional geomorphic and atmospheric controls on triggering events and runout behaviour. A large and long runout landslide on Wrangell Island, with area in the top 0.5% of >14,760 slides mapped in the Tongass National Forest, initiated during an atmospheric river event in November 2023 and travelled >1 km downslope, causing six fatalities. We used field observations, sequential airborne lidar, geotechnical analyses, and climate data to characterize the geomorphic, hydrologic, and atmospheric conditions contributing to the landslide. Rainfall intensities recorded at the Wrangell airport were modest (~1-yr recurrence interval), but rapid snowmelt and drainage from a ridgetop wetland may have contributed to rapid saturation of the landslide. Although strong winds were recorded, we did not observe extensive windthrow, which may downgrade its contribution to slope failure. The landslide mobilized a steep, thick (>4 m) wedge of colluvium that accumulated below a resistant bedrock ledge and entrained additional colluvial deposits as it travelled downslope across cliff-bench topography. The substantial entrainment resulted in an unusually large width, extensive runout, and low depositional slope as the landslide terminated in the coastal environment. Our results suggest that the sequencing of rain- and snow-dominated storms, geologic controls on post-glacial colluvium production and accumulation, and ridgetop hydrology contributed to landslide initiation and runout. Advances in post-glacial landscape evolution models that include colluvium production, frequent lidar acquisition, and additional climate data are needed to inform regional landslide hazard assessment.

## 1 Introduction

In steep, forested landscapes, shallow landslides serve as the primary agent of erosion (Hovius et al., 1997; Korup et al., 2010; Larsen et al., 2010; Swanson et al., 1987), produce and transport sediment that contributes to aquatic habitat (Geertsema and Pojar, 2007), set the relief structure of mountain ranges (Stock and Dietrich, 2003), and constitute a significant hazard to

proximal communities and infrastructure (Godt et al., 2022). In contrast to bedrock landslides whose failures are governed by bedrock properties (Schuster and Highland, 2001; Wyllie and Mah, 2004), shallow landslides composed of loose, unconsolidated material tend to initiate in zones of thick colluvium that experience variable saturation due to precipitation and snowmelt and in turn generate debris flows or debris slides with significant downslope runout and inundation (Gabet and Mudd, 2006; Iverson, 2000). In unglaciated terrain, these shallow landslides can initiate in a variety of landforms, but often occur in unchanneled valleys (or hollows) at the upstream tips of valley networks that are subject to cycles of infilling and excavation over $10^2$- to $10^4$-yr timescales (Benda and Dunne, 1997; Dietrich et al., 1986; D'Odorico and Fagherazzi, 2003). Characteristic ridge-valley sequences in these settings have facilitated the identification and characterization of shallow landslide and debris flow processes and informed models for soil transport, near-surface hydrologic response, and landslide initiation and runout (Dietrich et al., 1995; Lancaster et al., 2003; Montgomery et al., 1997; Reid et al., 2016; Schmidt et al., 2001).

In contrast, in steep, post-glacial settings, terrain morphology tends to be dominated by glacial landforms and deposits such that dissection is patchy and weakly established (Brardinoni and Hassan, 2006). Specifically, shallow landslides tend to initiate within soils of thin-to-moderate thickness (1-3 m) on steep planar slopes and runout to valley floors or low-order channels that are often highly unstable and subject to frequent reorganization (Brardinoni et al., 2009). In these highly dynamic settings, topographic controls on colluvium accumulation along steep, unchanneled slopes is poorly constrained making it difficult to predict landslide entrainment and volumetric growth which largely determine runout and inundation (Brien et al., 2025; Iverson and Ouyang, 2015; Patton et al., 2022). Furthermore, the relative importance of processes that generate the accumulation of colluvium, such as in-situ weathering of till or bedrock, transport of soil or talus deposits, and deposition of allochthonous deposits (e.g., tephra), and thus promote initiation and entrainment is also poorly known (Bovy et al., 2016; Spinola et al., 2024). In these formerly glaciated hillslopes, the lack of a conceptual framework for the production and transport of unconsolidated material inhibits our ability to identify areas susceptible to shallow landsliding, runout, and inundation (Brardinoni et al., 2018; Guthrie, 2002).

The triggering of shallow landslides and debris flows in post-glacial terrain is primarily accomplished by storm events that generate intense rainfall over several hours (Fan et al., 2020; Guthrie et al., 2010; Patton et al., 2023; Swanston, 1969) and shallow subsurface stormflow that saturates colluvium and leads to elevated pore pressures. For example, Patton et al. (2023) used logistic regression and Bayesian methods to demonstrate that 3-hr rainfall intensity can effectively differentiate storms that trigger debris flows near Sitka, Alaska. Their analysis forms the basis of the *sitkalandslide.org* warning system that uses National Weather Service (NWS) forecasts to define the 3-hr rainfall intensity with medium (7 mm hr$^{-1}$) and high (11 mm hr$^{-1}$) levels of risk up to 3 days in the future (Lempert et al., 2023). Additional studies also highlight the importance of rain-on-snow events that can rapidly advect large quantities of water into near surface soil and bedrock and contribute to slope instability (Darrow et al., 2022). Field observations from recently failed head scarps reveal evidence for abundant seepage

associated with permeability contrasts along the interface between bedrock, till, or colluvium as well as localized flow associated with fracture networks (Buma and Pawlik, 2021; Swanston, 1970). Notably, the upslope source of shallow groundwater and near-surface runoff that controls hydrologic response in post-glacial steeplands remains unclear owing to the paucity of high-resolution topography, field observations, and instrumental records in these settings. Additional factors contributing to landslide triggering in steep, forested terrain include timber harvest, fire, disease, and infestation, which can affect surface hydraulic properties and root reinforcement that resists sliding (Goetz et al., 2015; Johnson et al., 2000; Swanston, 1969). Extreme wind gusts have also been implicated in landslide initiation (Buma and Johnson, 2015; Lin et al., 2025; Parra et al., 2021) although the relative importance is difficult to determine and the mechanism by which wind gusts may contribute to slope failure is unclear and may include physical disturbance, progressive root failure, directed rainfall, or opening of hydrological pathways (Guthrie et al., 2010; Rulli et al., 2007). High winds can also contribute to rapid snowmelt (Hasebe and Kumekawa, 1995) and warm atmospheric rivers have been shown to promote snowmelt that substantially increases stream discharge (Guan et al., 2016; Hatchett, 2018; Henn et al., 2020).

The runout and inundation of landslides in post-glacial settings tends to be highly variable owing to variations in landscape dissection and the availability of unconsolidated material for entrainment. Most generally, the weakly-dissected slopes tend to generate fewer mobile slides that deposit on steeper slopes compared to slides in unglaciated settings (Booth et al., 2023; Vascik et al., 2021). Forest cover also affects landslide mobility, and the dense forest cover and large woody debris characteristic of many post-glacial settings favour lower mobility landslides (Booth et al., 2020). An abundance of datasets and models for the production and redistribution of colluvium that contribute to debris flow entrainment and volumetric growth have been generated in unglaciated settings (DiBiase et al., 2017; Gorr et al., 2022; Guilinger et al., 2023; Lamb et al., 2011; Reid et al., 2016; Rengers et al., 2020), but we lack both a framework and datasets that enable us to predict the runout, volume, and inundation of debris flows in post-glacial settings.

The need to improve our understanding of post-glacial shallow landslide triggers and processes in SE Alaska was highlighted by a large, catastrophic landslide that occurred on Wrangell Island on the evening of November 20, 2023. The mile point (MP) 11.2 landslide initiated during an intense rainfall event and entrained large quantities of colluvium and trees as it travelled downslope (Fig.1). Before terminating in Zimovia Strait, the MP11.2 landslide travelled over 1 km, inundated Zimovia Highway, destroyed three homes, and caused six fatalities (Nicolazzo et al., 2024), making it one of the deadliest landslides in Alaska history. This event was preceded and followed by several other fatal landslides in the region, including the 2015 Sitka, 2020 Haines, and 2024 Ketchikan events. This concentration of activity motivates a detailed assessment of the geomorphic, geologic, and atmospheric factors contributing to the initiation and runout of impactful landslides in SE Alaska. Here, we use an array of tools to characterize the 2023 Wrangell landslide and describe how these findings will advance our ability to assess landslides in the region. In particular, our analysis addresses: 1) atmospheric processes, including precipitation and wind, that contributed to event triggering, 2) controls on the accumulation of unconsolidated material that promotes landslide initiation,

downslope entrainment, and volumetric growth, 3) geologic and topographic controls on landslide runout and inundation, and
4) controls on the organization and evolution of upslope drainage networks that modulate hydrologic response at the initiation
zone. We leverage field observations, geotechnical measurements, sample analyses, climate data, change detection analysis
from sequential airborne lidar data, an existing US Forest Service landslide inventory, and hydrologic flow routing analyses
to assess the 2023 Wrangell landslide. Our findings provide critical information for identifying landslide triggers, mapping
susceptible initiation zones, and modelling runout and inundation, and we propose specific steps, and research needs to advance
landslide assessment in SE Alaska and other post-glacial landscapes to help reduce risk and minimize exposure during future
events.

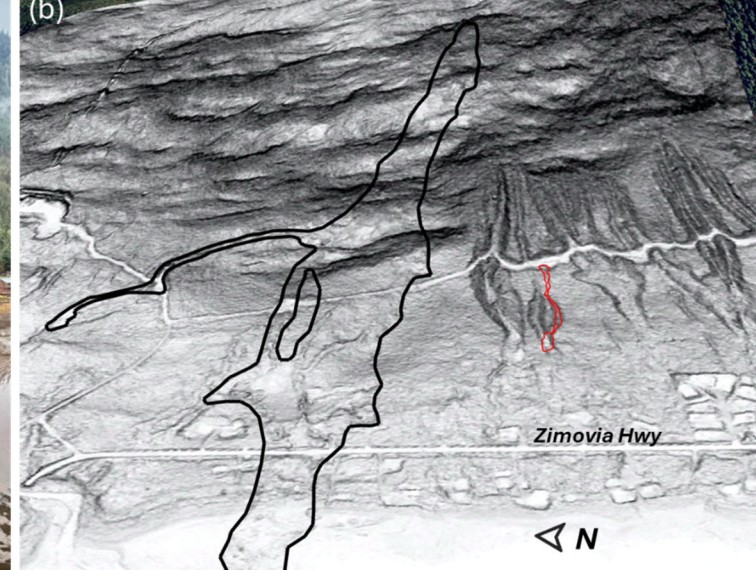

**Figure 1.** The November 20, 2023 MP11.2 Wrangell landslide, SE Alaska, USA: **(a)** Oblique aerial photograph facing east taken on
November 22, 2023 by B. Salisbury (DGGS), and **(b)** oblique lidar slopeshade image.

## 2 Study site

### 2.1 Geology and geomorphology

Situated in the southern half of southeast (SE) Alaska, Wrangell Island (Fig. 2) is composed of an assemblage of marine rocks
in the Gravina coastal belt on the eastern margin of the Alexander Terrane that composes a substantial fraction of the region
(Fig. 2b) (Haeussler, 1992; Wheeler and McFeely, 1991). Bedrock of the northern half of the island includes Cretaceous and
Jurassic graywacke and Cretaceous intrusions (Karl et al., 1999). These turbidites and igneous rocks were deformed in the
Late Cretaceous during the closing of a marine sedimentary basin between the Alexander terrane to the west and the Stikine
terrane to the east (Haeussler, 1992). The graywacke is part of the Seymour Canal Formation, a unit with fine-grained,
rhythmically bedded turbidite deposits that are regionally recrystallized to slate or phyllite. The sandstone layers tend to be
highly resistant and often form bedrock cliffs in areas with hillslope orientation that oppose dip direction.

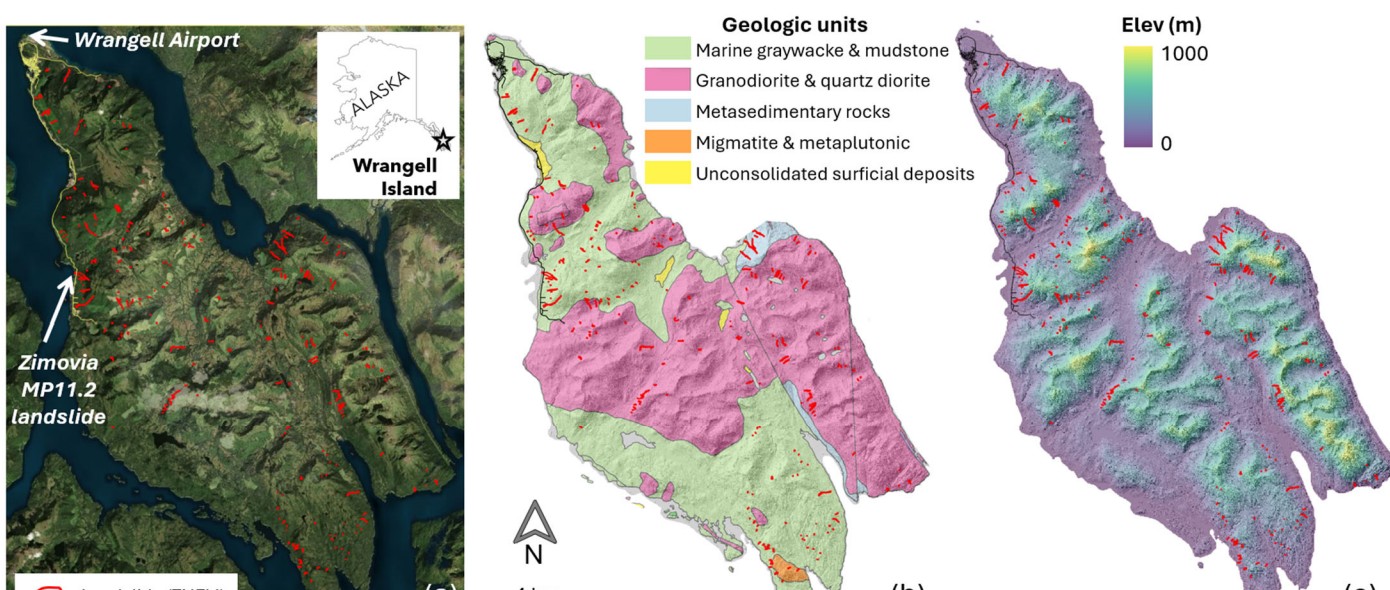

**Figure 2**. Maps of Wrangell Island overlain with 256 landslide polygons (red) from the Tongass National Forest Landslide
Inventory (TNFLI): (**a**) satellite imagery (Images © 2023 Planet Labs PBC)), (**b**) geologic units (Karl et al., 1999), and (**c**)
lidar elevation (Zechmann et al., 2023) and hillshade image.
The SE Alaska archipelago, including Wrangell Island, has been repeatedly glaciated, most recently during the Last Glacial
Maximum, generating characteristic landforms, including cirques, uplifted shorelines, and broad U-shaped valleys (Fig. 2c)
(Hamilton, 1994; Mann and Hamilton, 1995). By 13 to 15 kya, the margins of the Cordilleran Ice Sheet had retreated from SE
Alaska fjords, channels, and interior passages, leaving isolated or stranded ice caps on some islands, with alpine or tidewater
glaciers in many valleys and mountain peaks protruding above alpine glaciers (Carrara et al., 2003; Menounos et al., 2017).
Broad and gentle uplifted shorelines (sometimes more than 100 m above sea level) with beach ridges, storm berms, and weak
dissection, are abundant along coastlines in portions of SE Alaska (Baichtal et al., 2021) and may influence landslide runout.
On hillslopes, post-glacial landscape evolution is highly variable and some areas, particularly portions of western Wrangell
Island, experience widespread slope modification from rockfall, talus accumulation, localized gullying, and landsliding.

## 2.2 Climate and vegetation

SE Alaska is a regional temperate rain forest with a maritime climate (Wendler et al., 2016). In Wrangell the mean annual
precipitation is roughly 2 m, most of which falls as rain at low elevation with the proportion of rain-to-snow decreasing with
elevation. In Wrangell and across SE Alaska, nearly all high-intensity rainstorms are associated with atmospheric rivers (ARs)
(Nash et al., 2024), which are long (>2000 km), narrow (<500 km), moisture-laden currents in the lower troposphere (Neiman
et al., 2008; Ralph et al., 2004). When ARs, which are most active August to November in SE Alaska, make landfall, orographic
forcing can result in higher precipitation in mid-slope locations and on slope aspects that coincide with the trajectory of
incoming ARs (Marra et al., 2022; Rulli et al., 2007). Although ARs account for only ~33% of annual precipitation, they
generate 90% of extreme precipitation in the region (Sharma and Déry, 2020). As a result, ARs trigger the vast majority of
shallow landslides along the Pacific coast of North America and SE Alaska, although these slide-triggering ARs are a small
fraction of all ARs that make landfall (Cordeira et al., 2019; Oakley et al., 2018).
Much of SE Alaska is densely forested with mixed conifer forests of western hemlock (*Tsuga heterophylla*), Sitka spruce
(*Picea sitchensis*), western red cedar (*Thuja plicata*), yellow cedar (*Callitropsis nootkatensis*), and mountain hemlock (*Tsuga*
*mertensiana*) (Harris and Farr, 1974; Hees and Mead, 2005). Disturbed and riparian areas host locally abundant red alder and
black cottonwood. Non-forested regions include high-elevation tundra vegetation and emergent wetlands (e.g., muskeg),
surface water, glaciers, and snow/icefields (Flagstad et al., 2018). On Wrangell Island, logging since the 1950's along lower
elevations has resulted in a mosaic of forest stand age. Although recent hemlock sawfly and western blackheaded budworm
outbreaks have resulted in swaths of mid-elevation trees that have dropped their needles (Howe et al., 2024), the extent of tree
mortality and impact on root systems, and thus slope stability, is not yet established.

## 2.3 Landslides in Southeast Alaska

Based on the Tongass National Forest Landslide Inventory (TNFLI), which includes >20,000 mapped slope failures and slide-
prone areas (U.S. Forest Service, 2025b), the vast majority (>80%) of landslides in SE Alaska are debris flows or unchannelized
debris avalanches that initiate within weathered till or colluvium during periods of intense rainfall (Fig 2a). The recent fatal
landslides in SE Alaska were colluvial landslides, except for the 2020 Beach Road landslide in Haines that initiated within
shallow bedrock during the December 2020 rain-on-snow event (Darrow et al., 2022). Extensive field-based research on
landslide processes, particularly root reinforcement and hydrologic response, originated in the 1960s on Prince of Wales Island
following increased landslide activity after timber harvest (Johnson et al., 2000; Swanston, 1969, 1970, 1973). These studies
indicated that tree mortality affected landslide density as well as runout, such that landslides in harvested areas exhibited higher
mobility (Booth et al., 2020; Buma and Johnson, 2015). The wide glacial valleys and weakly-dissected slopes in SE Alaska
tend to favour infrequent landslide delivery to streams and most debris flow deposits contribute to fans or footslope deposits.

**2.4 The November 20, 2023 atmospheric river and impacts on Wrangell Island**

A hurricane-force 964 mb low pressure system lifted out of the North Pacific into the Gulf of Alaska during the early morning hours of November 20, 2023 (Figure 3a). This low-pressure system proceeded along a north-northwest track, with the warm front moving over southern and central SE Alaska before the front pushed north through the evening hours (Fig. 3b). A cold air mass over northern SE Alaska and the Yukon produced a zone of high pressure and a strong pressure gradient across SE Alaska. This colder air likely produced snowfall at higher elevations prior to the arrival of warm, moist air. This weather system included significant subtropical moisture and additional AR characteristics evident in satellite imagery (Fig. 3c). At 3 PM on November 20, the CIMMS Model analysis of Integrated Water Vapor Transport (IVT), a commonly used indicator of ARs, indicated very high IVT over the southern half of SE Alaska (Fig. 3d).

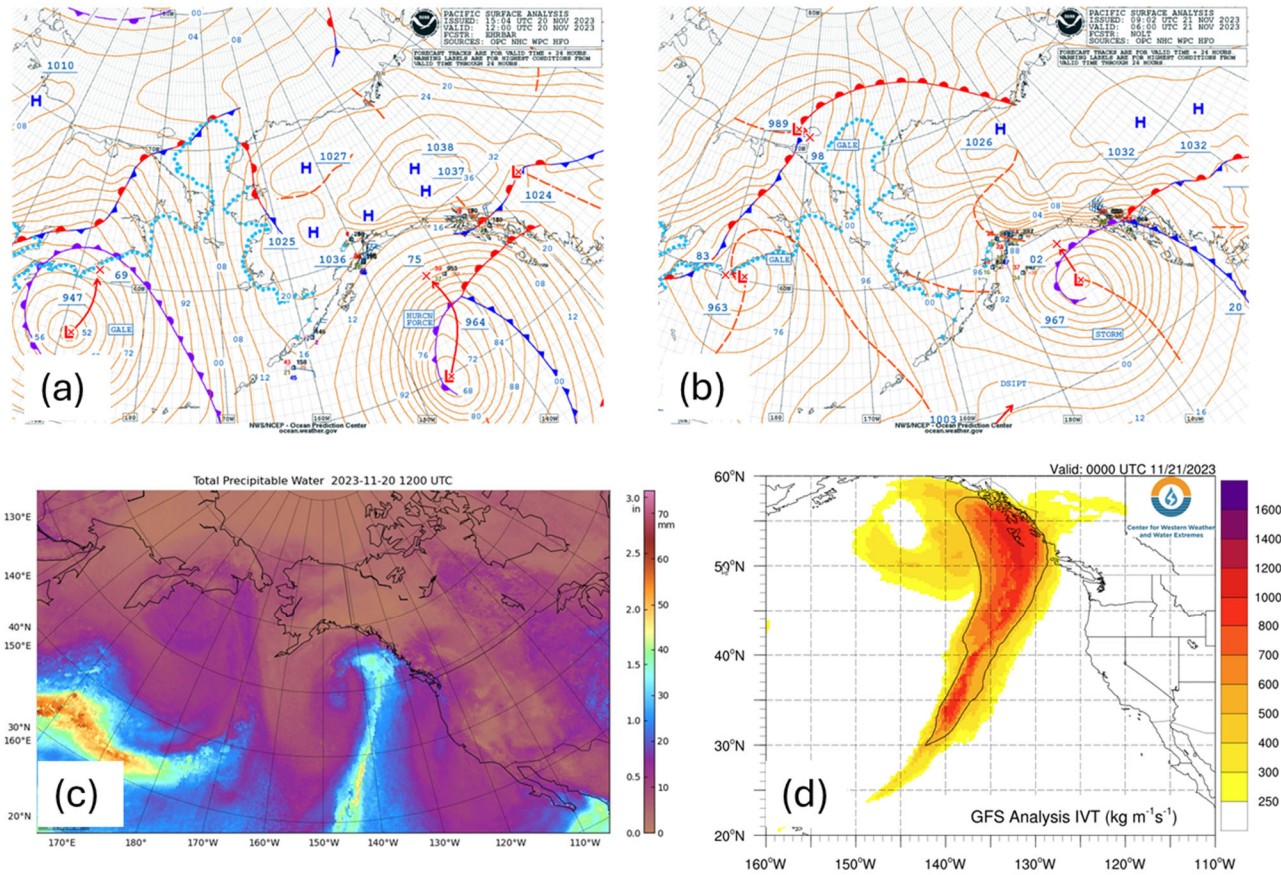

**Figure 3.** The November 20, 2023 atmospheric river event that triggered the MP11.2 landslide: (**a**) NOAA surface analysis from imagery from November 20 at 4am Alaska Standard Time (AKST), (**b**) NOAA surface analysis from imagery from November 20 at 10pm AKST, (**c**) Total Precipitable Water from the Cooperative Institute for Meteorological Satellite Studies (CIMSS) Morphed Integrated Microwave Total Precipitable Water (MIMIC-TPW) for November 20 at 4am AKST, (**d**)

Integrated Water Vapor Transport (IVT) from Center for Western Weather and Water Extremes (C3WE) from November 20
at 4pm.

Heavy precipitation and high wind gusts began in the morning hours of November 20 and warm air and moisture combined
with high winds likely melted snow at higher elevations. That afternoon, numerous landslides and road blockages were reported
on Prince of Wales Island near Craig, Klawock, and Black Bear. The front shifted to an eastward trajectory in the early evening
hours, as heavy rain and winds shifted towards Wrangell Island, and reports of the Zimovia Highway MP11.2 landslide were
received just before 9 pm Alaska Standard Time. The front continued eastward, and rain and winds diminished through the
night. The 24-hr precipitation totals on Prince of Wales varied from <5 cm to >16 cm on the east and west sides of the island,
respectively (National Oceanographic and Atmospheric Administration (NOAA), 2024). At Wrangell airport, which is situated
at sea level near the northern tip of the island and over 15 km north of the MP11.2 landslide (Fig. 2a), 8 cm of rainfall was
recorded in 24 hours, and nearly half of that rainfall total was delivered steadily between 3 pm and 9 pm (Fig. 4). Peak wind
speed and gusts of 30 and 50 km hr$^{-1}$, respectively, occurred from 11 am to 3 pm and sustained at high levels through the
evening.  Air temperature rose rapidly in the morning and remained above 5°C. A remote weather station located ~25 km west
of the MP11.2 slide at 275m above sea level on Zarembo Island recorded similar wind speeds as the Wrangell airport but
notably logged a short period of gusts >100 km hr$^{-1}$ around 7 pm in conjunction with a southward shift in direction of the front
(Nicolazzo et al., 2024). Local observations during the day of the storm are notable because several residents reported: 1)
rainfall to be more intense along Zimovia Highway than in Wrangell, and 2) significant snow cover at mid-to-high elevations
coincident with the initiation zone prior to the November 20 storm that melted by November 21.

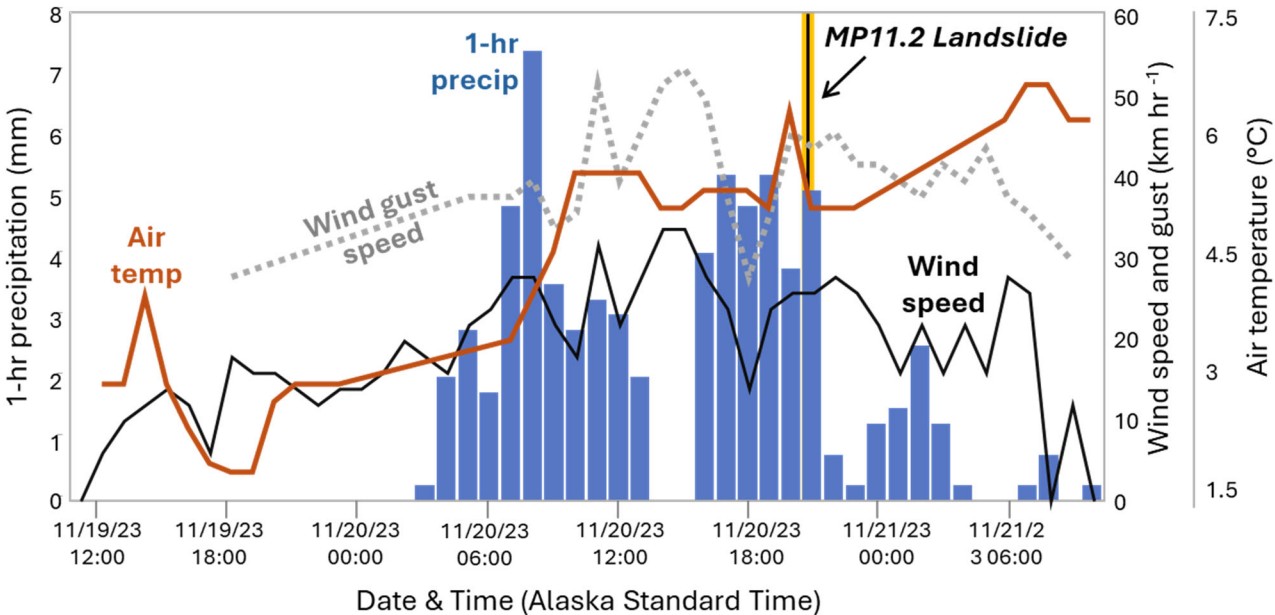

**Figure 4.** Time series of hourly climate data from Wrangell Airport spanning November 19 to 21, 2023, including: 1-hr precipitation (blue bars), average wind speed and maximum wind gusts (black and dashed grey lines, respectively), and air temperature (red line). The MP11.2 landslide occurred at ~9pm on November 20 (vertical yellow/black line).

On November 21, Alaska Governor M. Dunleavy issued a state disaster declaration, and the Alaska Division of Geological & Geophysical Surveys (DGGS) was contracted to document landslides triggered on Wrangell Island during the storm. DGGS used airborne lidar acquired in July and on November 28-29, 2023, to estimate the character and volume of the MP11.2 and nearby landslides (Nicolazzo et al., 2024). For the MP11.2 landslide, they noted about 80,000 $m^3$ of erosion, thick soil entrained along bedrock benches, and an abundance of soil and large woody debris (with a negligible amount of bedrock) composing the deposit. Portions of the deposit had been removed before the post-event lidar acquisition and the deposit travelled nearly 150 m into the ocean, such that a small but non-negligible fraction (<15%) of the deposit was not captured with lidar differencing.

## 3 Methods

### 3.1 Overview

To assess and quantify controls on the initiation and runout of the MP11.2 landslide, we performed a wide array of analyses and generated observations from fieldwork, community events, airborne lidar, hydrologic modelling, weather data, and

geotechnical testing. We endeavoured to address community-generated queries, such as the potential role of wind as a
triggering agent and mechanisms responsible for the anomalously large size of the landslide.
**3.2 Landslide geometry**
To contextualize the MP11.2 landslide, we analysed the landslides previously mapped on Wrangell Island included in the
TNFLI (n=20,235) (U.S. Forest Service, 2025b). We excluded snow avalanche chutes, snow avalanche fields, and debris
avalanche fields because these extensive features reflect landforms that accumulate over time rather than discrete landslide
events. For the remaining landslides (n=14,670), we identified those occurring on Wrangell Island and quantified the area,
mobility (defined as H/L, where H is elevation difference between the head scarp and deposit and L is landslide length, defined
as the horizontal distance between the head scarp and end of deposit), and aspect ratio (defined as W/L, where W is average
landslide width).
**3.3 Field observations, sampling, and analyses**
To document failure mechanisms and runout behaviour, we traversed the entire length of the landslide, observing evidence of
entrainment and deposition, and mapping localized seepage in the head scarp area. We collected representative soil samples,
from which we determined gravimetric water content (American Society for Testing Materials, 2017a); particle-size
distribution, consisting of sieve analysis (American Society for Testing Materials, 2017b), sedimentation analysis (American
Society for Testing Materials, 2021), and specific gravity testing (American Society for Testing Materials, 2014b); Atterberg
limits (American Society for Testing Materials, 2017a), and organic content by loss on ignition (Alaska Department of
Transportation and Public Facilities, 2023). We also collected volumetric samples using a soil sampler with inner brass rings,
from which we determined dry unit weight and volumetric water content. We classified samples using the Unified Soil
Classification System (American Society for Testing Materials, 2017a). We also collected estimates of intact bedrock strength
using two Rock Schmidt Rebound Hammers (N-type and L-type, with impact energies of 2.207 Nm and 0.735 Nm,
respectively). We followed standard methods (American Society for Testing Materials, 2014a) with the exception that we did
not use a grinding stone on the in-situ rock faces. We also collected slices (or "cookies") of four trees entrained in the deposit
to determine their ages and obtained 35 bedrock and/or joint surface orientation measurements for kinematic analysis of sliding,
wedge, and toppling failure. Finally, we ventured onto the ridgetop above the landslide to document the upslope accumulation
area that contributes surface water flow to the head scarp region.
**3.4 Geospatial analysis: change detection, morphology, and hydrologic modelling**
To quantify the pattern of erosion and deposition, and controls on colluvial deposits and their entrainment in the landslide, we
used the July 2023 and November 2023 lidar for change detection and topographic analysis (Zechmann et al., 2023, 2024).
Both datasets have 0.5-m pixel spacing, high bare earth point density (>5 pts $m^{-2}$), and high accuracy (<10 cm error in bare
and vegetated areas). We used QGIS for our analyses and mapped the landslide extent using high-resolution optical imagery
acquired by the Alaska Department of Transportation and Public Facilities (ADOT&PF) and the airborne lidar data. By
comparing numerous stable features in both lidar datasets we determined systematic offset to be negligible (<3 cm). For change
detection, we applied raster-based subtractions and created a point layer for the landslide pixels, which we attributed with
slope, elevation, land surface change using the digital terrain model (DTM, i.e., bare earth data), and vegetation change using
the digital surface model (DSM, i.e., first return data). We used the derived points and their attributes in three primary ways:
1) maps of DTM and DSM change across the landslide and surrounding terrain, 2) plots of swath (10-m wide) averaged profiles
of elevation, slope, and DTM / DSM change along a longitudinal transect that spans the central axis of the primary landslide,
and 3) construction of a comprehensive mass balance of DTM change (i.e., erosion and deposition) along a cross-sectional
transect that spans the entire width of the landslide.
For the hydrologic modelling, we used TopoToolbox to define flow paths above the landslide scarp by removing sinks and
defining flow directions and flow accumulation using a multiple flow direction (MFD) algorithm that partitions flow to all
downslope pixels in proportion to the gradient of each pixel (Schwanghart and Scherler, 2014). In addition, we accessed the
U.S. National Wetlands Inventory (Flagstad et al., 2018) in conjunction with our flow model to assess the potential contribution
of wetlands to surface water flow and landslide triggering.

## 253 4 Results

### 254 4.1 Landslide geometry

The MP11.2 landslide has an area greater than 142,000 $m^2$ and initiated at 454m above sea level (as defined by the head scarp)
before flowing downslope >1km and depositing into the coastal marine environment (Fig. 5a). Although the width of the
landslide averages 130 m, it is widest in the middle of the runout zone, and relatively narrow (<50m) at the initiation zone and
terminus. Our analysis of landslides on Wrangell Island and in the TNFLI demonstrates that the MP11.2 landslide is notable
for its areal extent (Fig. 6a), which is more than twice the size of the next largest Wrangell Island landslide. When compared
to the entire TNFLI, the MP11.2 landslide has a larger area than 99.5% of the landslides (Fig. 6b), which further demonstrates
its exceptional size.
Given that landslide mobility (quantified as H/L, the value of which decreases with increased mobility) tends to vary with
landslide size (Corominas, 1996; Iverson et al., 2015; Rickenmann, 1999), we plotted H/L versus landslide area for the
Wrangell Island landslides and fitted a logarithmic trend, such that H/L decreases slightly with area (Fig. 7a). In this context,
the MP11.2 landslide is situated on the trend and thus does not appear notable for its mobility relative to its area. Because the
MP11.2 landslide maintained a relatively wide footprint along most of its path, we also plotted W/L versus area (Fig. 7b) and
noted a robust power-law trend indicating that slides tend to become increasingly elongate as they get bigger. In this context,
the MP11.2 landslide is anomalous for its large W/L value relative to its area. Specifically, the landslide plots well above the
trend and only one of the 25 next largest landslides has a similar positive deviation above the area-W/L curve (Fig. 7b). In
summary, the landslide did not appear to exhibit uncommon mobility as defined by H/L values, but rather it attained a large
area while also maintaining substantial width, which contributed to its extensive inundation area and devastating impact.

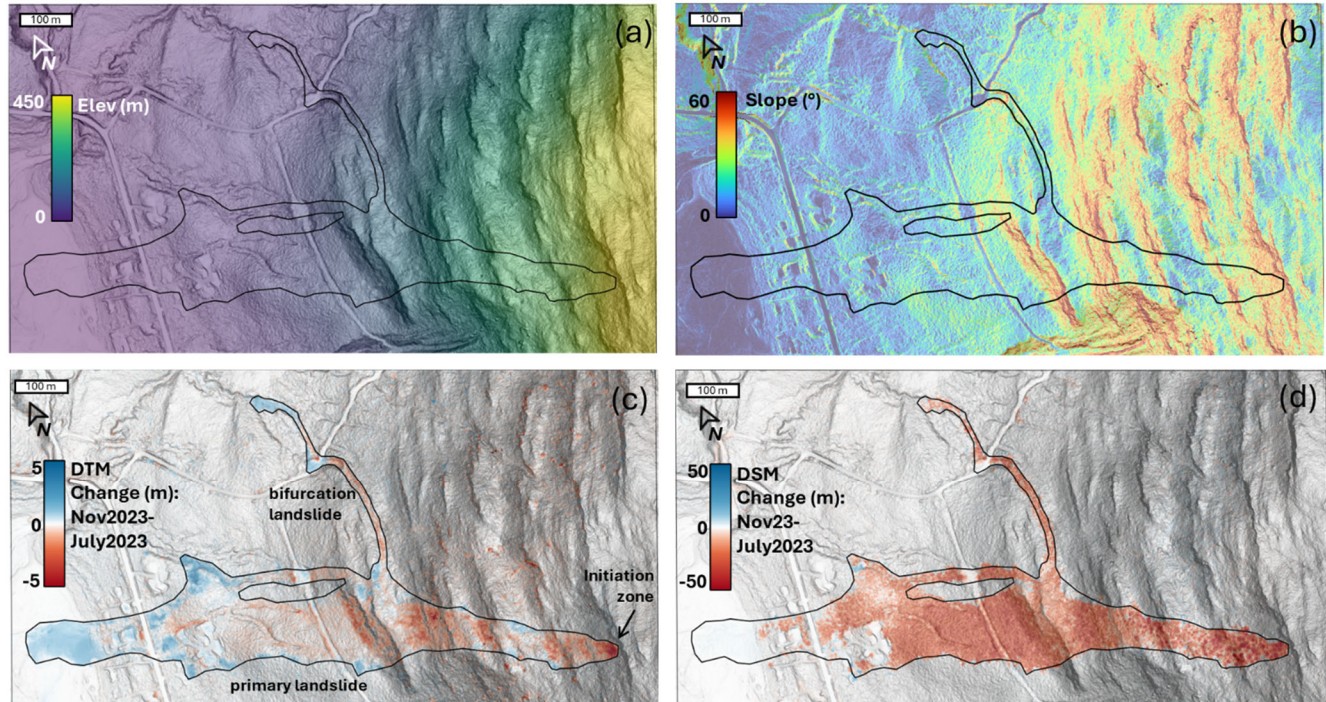

**Figure 5.** Lidar-derived maps (0.5-meter grid spacing) of the MP11.2 landslide: (**a**) elevation above sea level in meters, (**b**)
slope in degrees, (**c**) DTM change (land surface or bare earth), and (**d**) DSM change (first return or canopy) with November
2023 dataset subtracted from the July 2023 dataset such that negative values (red) reflect decreases and positive values (blue)
reflect increases.


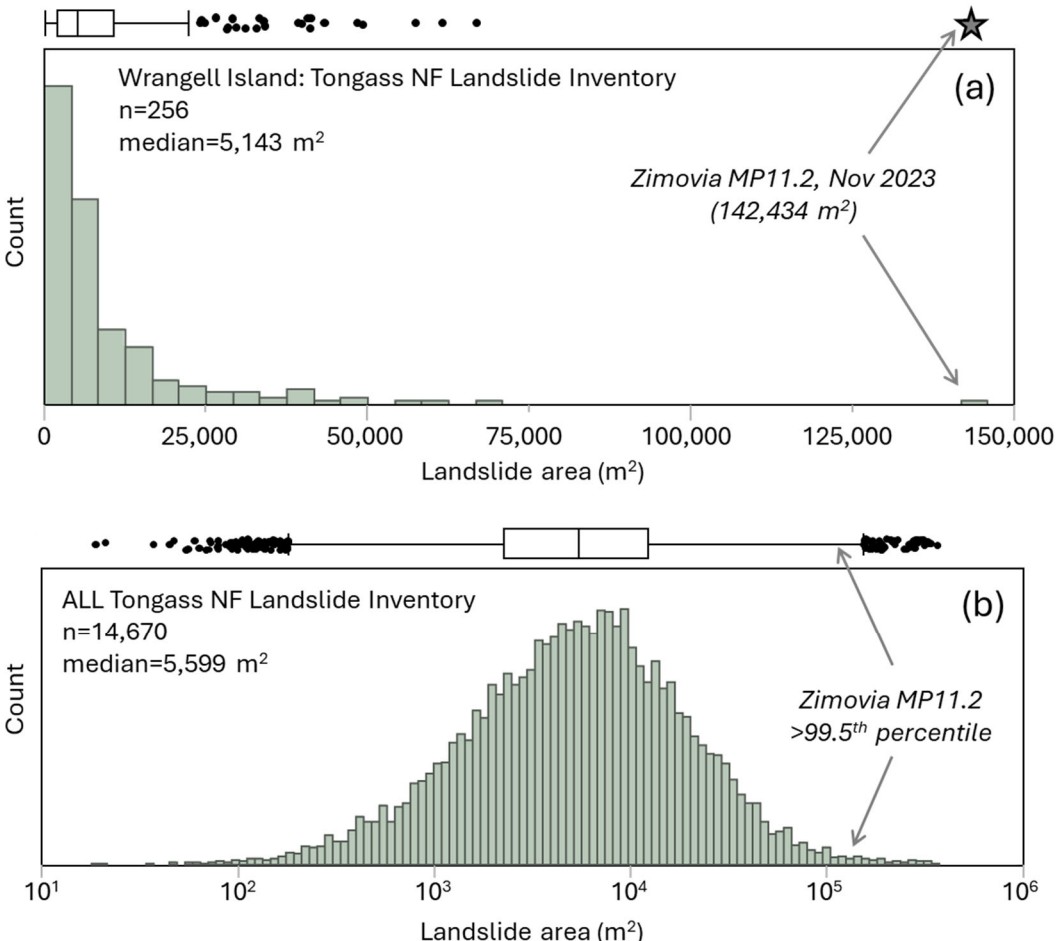

Figure 6. Histograms of landslide area from the TNFLI for (a) Wrangell Island and (b) all of the TNLFI, note the log scale. The box-whisker plots above each histogram convey the median, interquartile range and outliers and the star denotes the MP11.2 landslide.

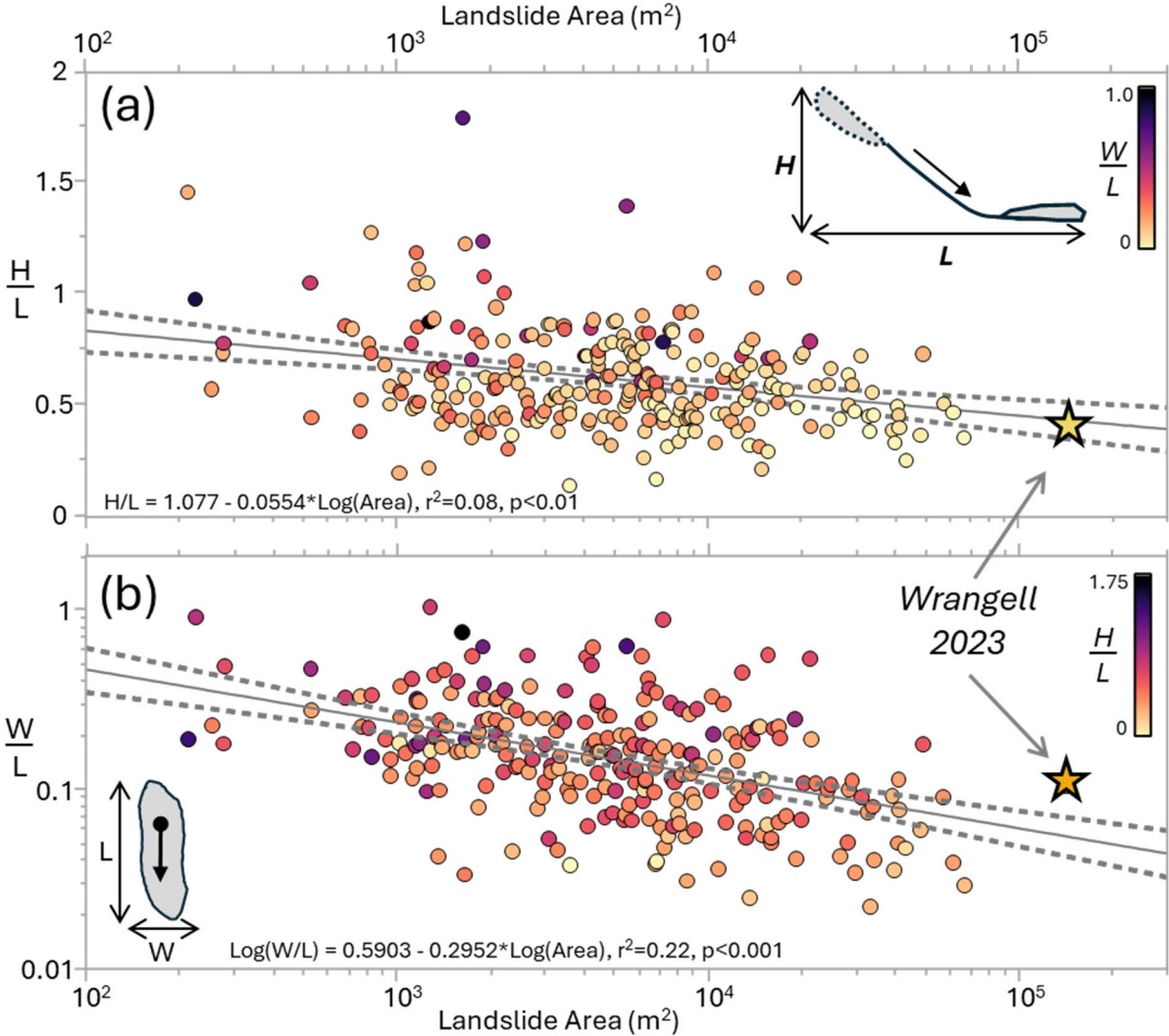

**Figure 7**. Plots of landslide characteristics for Wrangell landslides in the TNFLI. (**a**) Variation of mobility (H/L) with area, and (**b**) variation of landslide aspect ratio, defined as the ratio of width to length (W/L), with area. Note that the star indicates the MP 11.2 landslide in a and b. The solid black and dashed grey lines denote the regression fits and 95% confidence intervals for the equations indicated in a and b. Individual points are coloured by W/L in a and H/L in b.

## 4.2 Geologic units, bedrock structure, and soil properties

Marine sedimentary rocks of the Seymour Canal Formation (Karl et al., 1999) are exposed in the landslide and a bedrock quarry approximately 1.5 km north of the landslide head scarp (NE corner of map in Fig. 8). Bedrock lithology includes interbedded shale and graywacke typical of turbidite sequences with bedding dipping into the hillslope (to the east) within the landslide (Fig. 8). Local metasedimentary rocks on nearby hillslopes (slate and minor phyllite) indicate low-grade

metamorphism in the study area. Graywacke beds are 0.25- to 5-m thick as observed in the field and form benchlike
topography, with the resistant graywacke creating subvertical cliff bands within the landslide margin and across undisturbed
hillslopes, and the relatively weak shale forming low-gradient slopes (Fig. 5b). Bedding orientation in the quarry dips to the
southeast, indicating hillslope-scale folding (Fig. 8). In addition to bedding geometry, we documented three joint sets to assess
the potential for rock slope instability along the resistant bedrock cliffs. Preliminary kinematic analysis of discontinuities using
conservative friction angle estimates of 15° and 30° for shale and sandstone, respectively (Gonzalez de Vallejo and Ferrer,
2011), indicates that flexural toppling is possible while other rock failure mechanisms (direct toppling, wedge and planar
failure) are unlikely (supplemental materials).

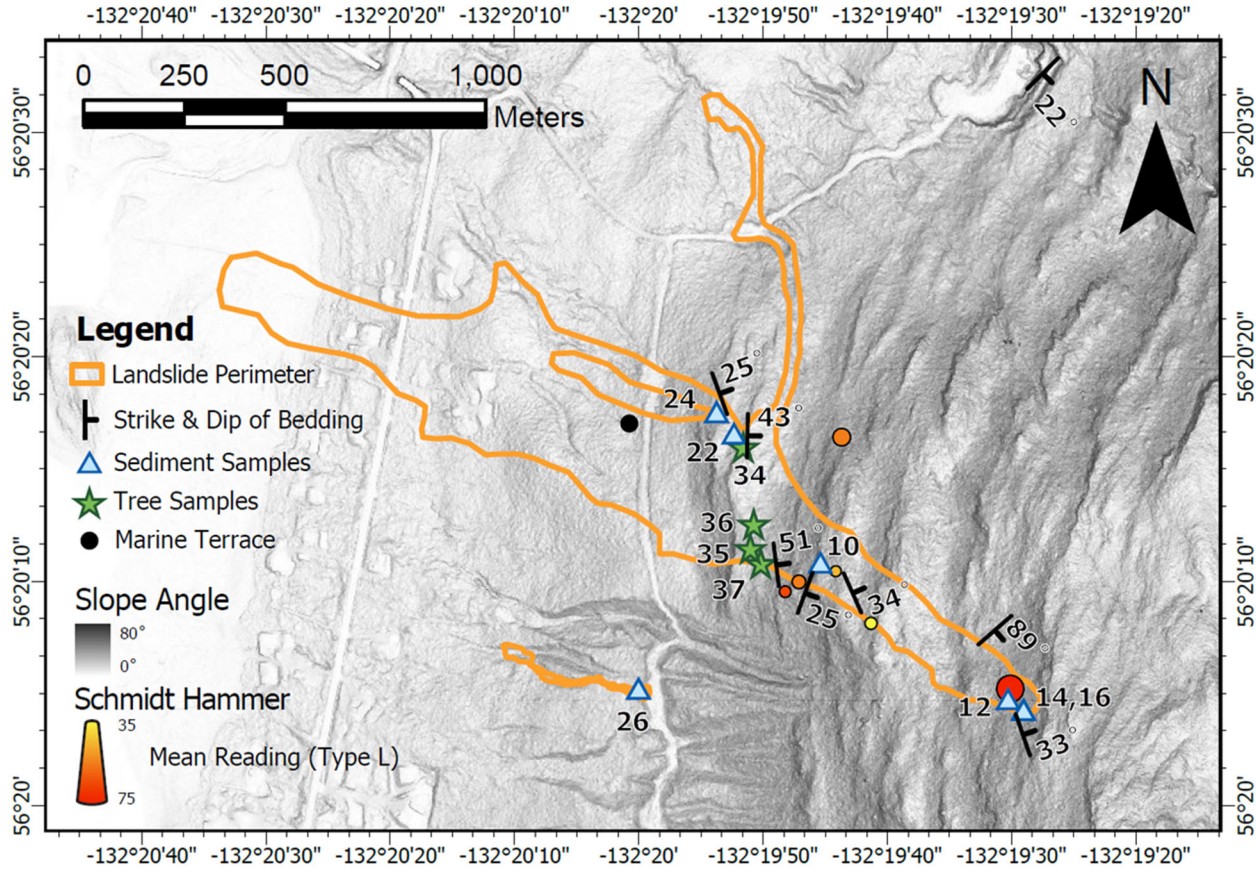

**Figure 8.** Lidar hillshade map of MP11.2 landslide showing locations of field measurements and samples acquired during the
August 2024 field campaign. Strike and dip and Schmidt Hammer values denote averages within each sample locale.
We collected a total of 60 readings with each of the Schmidt Hammers. Using a correlation for sandstone, siltstone, and
mudstone that does not require rock density and uses the L-type hammer (Aydin and Basu, 2005), our estimates of uniaxial

compressive strength (UCS) average 90 MPa for measurements taken outside the lateral margins of the landslide body, 82 MPa for measurements in the middle of the landslide body, and 148 MPa for measurements taken on massive greywacke exposed in the head scarp (supplemental materials). These values are typical for graywacke (Gonzalez de Vallejo and Ferrer, 2011) and indicate that the estimated UCS of the exposed graywacke in the head scarp is 80% higher than that within the lower landslide body and 64% higher than bedrock exposures adjacent to the MP11.2 landslide.

In exposures along the landslide flanks, we observed colluvium as discontinuous "wedges" at the base of bedrock cliffs, including a ~4 m thick deposit that constitutes the initiation zone (Fig. 9a). The matrix of the colluvium was brown, organic silty sand to silty sand with gravel (SM), similar to displaced landslide material observed downslope. The material properties of the colluvium imply moderate frictional strength, minimal weathering or alteration, and relatively high permeability. All of the samples tested were non-plastic (supplementary materials). In an area scoured by the landslide in its depositional zone, we also observed a deposit of sand and subrounded, imbricated gravel characteristic of coastal marine sediments. The deposit is exposed just below the USFS road at approximately 100 m elevation, which is consistent with the elevation of glacial isostatic adjustment documented for the region (Baichtal et al., 2021).

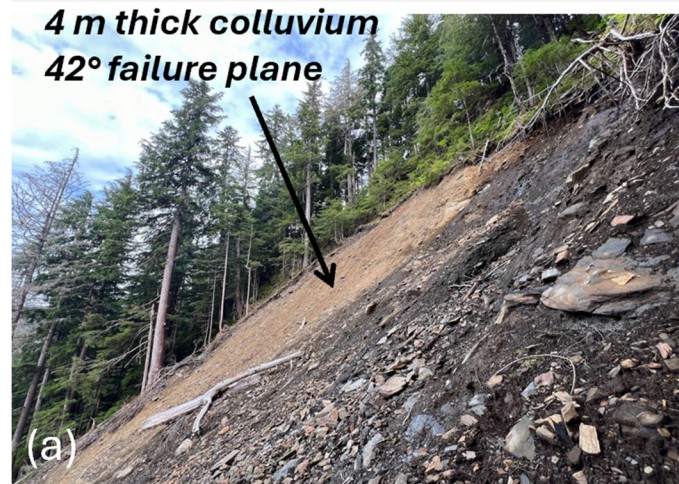

*4 m thick colluvium*
*42° failure plane*

(a)

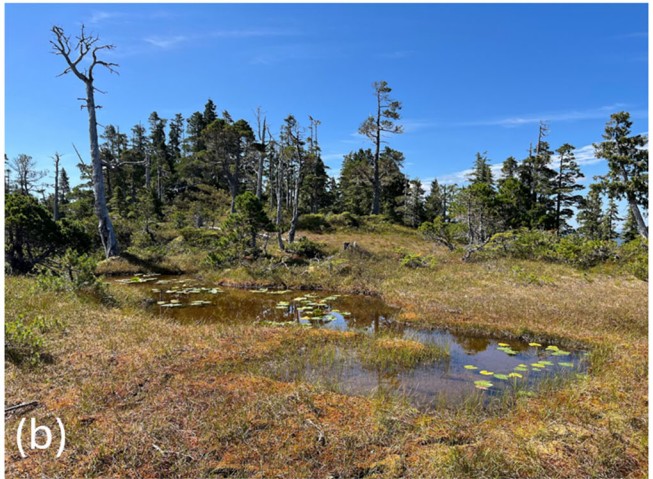

(b)

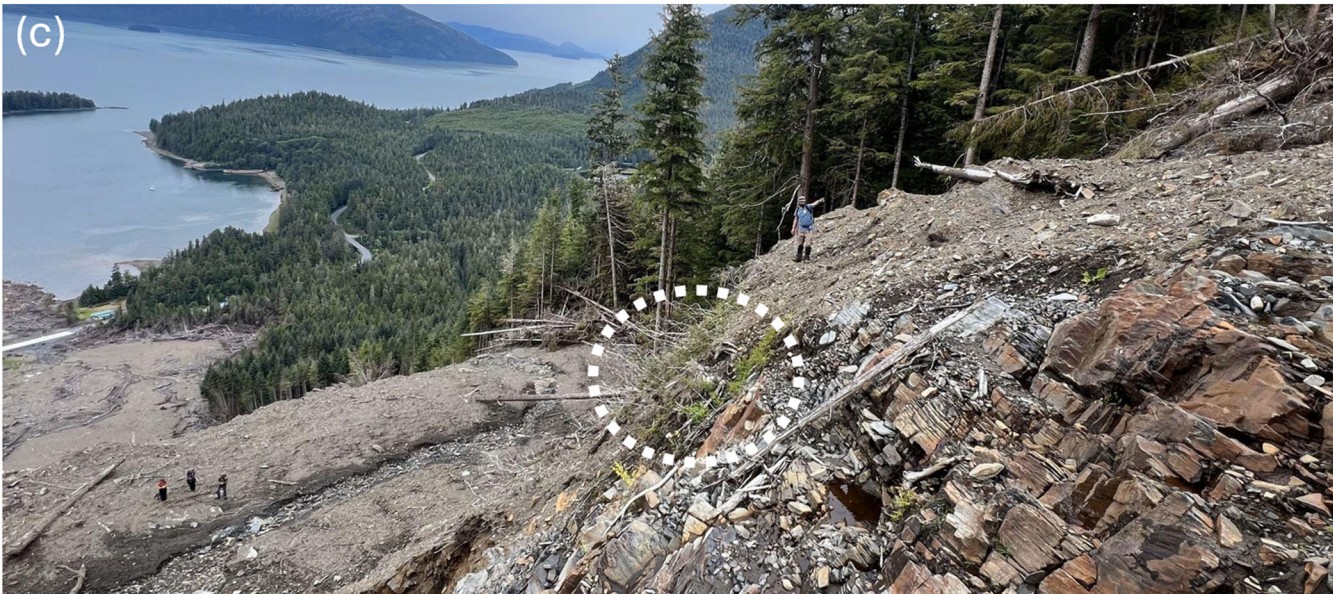

(c)

**Figure 9.** Photographs of key features identified in the field: (**a**) view to the north across the head scarp, exposing thick colluvial wedge in lateral margin, (**b**) ridgetop wetland or muskeg that drains to the head scarp, and (**c**) oblique view of mid-slope location (~1,000 m on transect; see Fig. 11) depicting high relief and resistant cliff-forming unit and patch of live blueberry bushes denoted by dashed white circle just below the top of the bedrock cliff. Note person for scale.

## 4.3 Landslide initiation and triggering factors

The initiation zone for the MP11.2 landslide has an average slope of 42±2.5° and is approximately 30-m wide and 26-m long (Fig. 5b, d). According to lidar differencing of the pre- and post-event DTMs, the average thickness of the initiation zone is 4.5±0.7 m (Fig. 5c), which is thick relative to many landslides observed in the Tongass National Forest (U.S. Forest Service,

2025b). In the days following the landslide, aerial imagery acquired by the Alaska Department of Transportation and Public
Facilities (ADOT&PF) revealed prodigious seepage emanating from the SE corner of the head scarp, and during our August
2024 field campaign we noted localized seepage in that location despite negligible rainfall in the preceding days. Additional
triggering factors include compromised root reinforcement, and we noted an abundance of standing Western hemlock trees
without needles just beyond the northern and southern margins of the initiation zone.
In the 6 hours prior to the MP11.2 landslide, rainfall intensity at the airport averaged 5 mm hr$^{-1}$ (Fig. 4), which corresponds to
a ~1-yr return interval (National Oceanographic and Atmospheric Administration (NOAA), 2024). In addition, the maximum
3-hr intensity just prior to the slope failure was less than the 7 mm hr$^{-1}$ intensity threshold that delineates an elevated level of
risk in the Sitka region (Patton et al., 2023). Notably, high winds and warm temperatures characterized the 12-hour period
prior to the landslide, and these changes may have contributed to the failure through mechanical disturbance and rapid delivery
of snowmelt to the initiation zone. Observational records of these potential triggering factors proximal to the landslide are
lacking, so we explored alternative sources of evidence. To assess the potential role of wind disturbance in landslide triggering
we used differencing of the canopy (or DSM) lidar data to map wind throw (or tree turnover) as a signature of canopy
disturbance proximal to the initiation zone (Fig. 5d). Consistent with our field observations, our map of DSM change does not
reveal evidence for widespread canopy disturbance beyond the margins of the landslide. In fact, the DSM change map revealed
less than 10 individual and localized tree turnover events dispersed within several kilometres of the MP11.2 landslide.

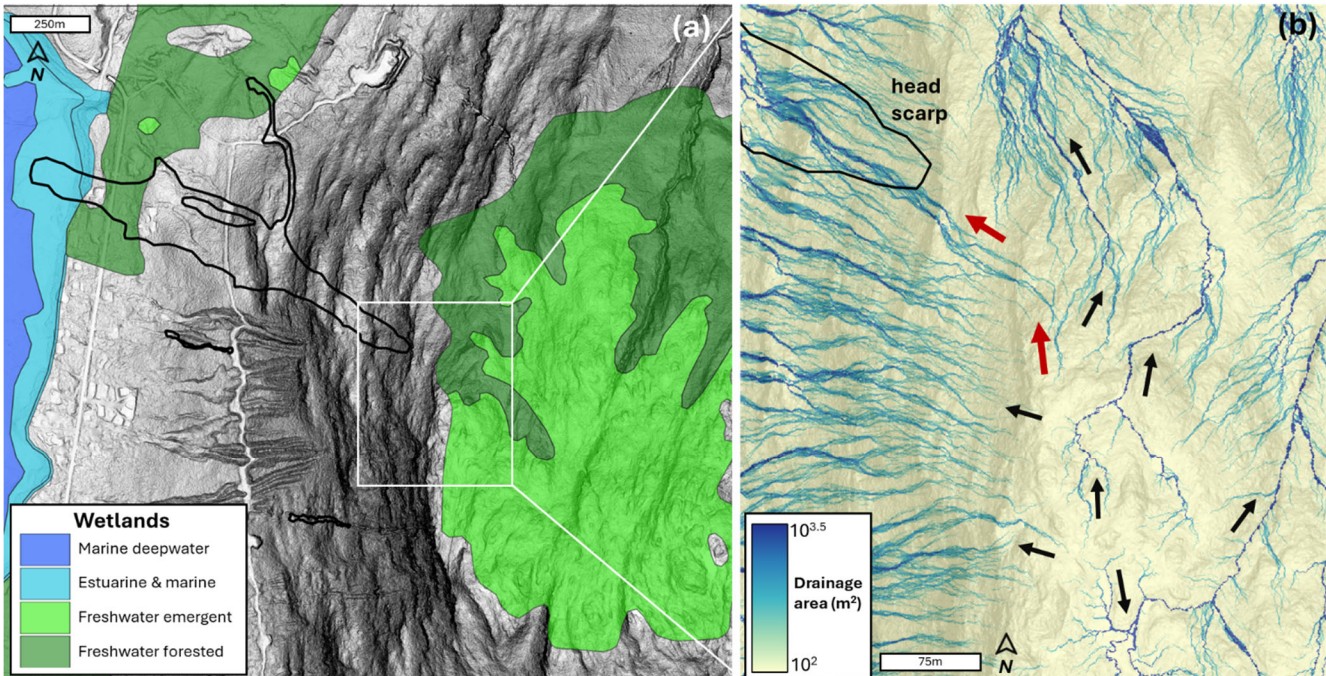

**Figure 10.** Lidar maps of MP11.2 landslide and relevant drainage features. (**a**) Shaded relief image of landslide and extent of
ridgetop wetland from the National Wetlands Inventory, (**b**) map of contributing drainage area along the ridgetop above the
MP11.2 landslide. Note the radial flow pattern that includes a significant area that contributes flow to the head scarp denoted
by the red arrows. The broad light blue flow lines reflect diffusive, unchannelized flow, while the narrow, dark blue flow
lines result from well-defined channels as seen by the black arrows in the eastern half of the image.

To assess the potential contribution from ridgetop wetlands and rapid snowmelt on the saturation of the initiation zone, we
mapped wetlands and hydrological flow paths upslope of the head scarp (Fig. 10a). Our map shows a radial drainage pattern
emanating from the ridgetop with an array of dispersed, west-directed flowpaths that drain to slide-prone slopes to the south
of the MP11.2 landslide (Fig. 10b). By contrast, flowpaths oriented to the north, east, and south tend to exhibit an incised and
well-defined channel network structure that is reflected by the narrow, dark blue (high drainage area) tendrils that contrast
with the more diffusive flowpaths with wider and lighter blue (lower drainage area) signatures draining west (Fig. 10b). This
pattern likely reflects the relative antiquity of channels and flowpaths draining from the ridgetop to the north, east, and south.
Notably, an elongate system of flowpaths is situated between the west- and north-directed drainages. This flow accumulation
pathway denoted by red arrows in Fig. 10b demarcates a substantial drainage area directed to the SE corner of the MP11.2
landslide head scarp and coincident with abundant seepage observed in the field. Our flow mapping indicates greater than
6,000 m$^2$ of drainage area upslope of the head scarp, and this source area includes a substantial fraction of low-gradient,
emergent wetlands with patchy bedrock exposure (Fig. 9b, 10a). In the field, this ridgetop wetland area (muskeg) was
characterized by deep (>2 m), organic soil akin to peatlands. Curiously, the flowpaths that contribute to the landslide head
scarp also reveal evidence of bifurcation into slide- and north-directed drainage systems (Fig. 10b). Our field observations
indicate that this bifurcation corresponds to meter-scale roughness in the bedrock/wetland surface, implying that the orientation
of ridgetop drainage may be highly dynamic and sensitive to local disturbances.
**4.4 Landslide runout and mass balance**
Our lidar and field analyses reveal strong topographic and geologic controls on the pattern of erosion and deposition along the
landslide runout path (Fig. 11). These analyses focus on the primary landslide path and do not include the north-directed
bifurcation that occurred in the middle sections and accounted for a small fraction (<10%) of the slide volume. Our field
observations indicate that the initiation zone was localized to the upper 30 m (~1350 m on our transect; Fig. 11a) such that
runout processes are responsible for the downslope pattern of erosion and deposition. The W-NW directed path of the slide
does not exhibit topographic convergence as expressed by contour (or planform) curvature and thus lateral confinement did
not affect the runout behaviour. Rather, our 10-m wide swath-averaged transect data show that the lower half of the ~1,250-m
long runout is characterized by a low-gradient surface with slope angles that seldom exceed 20° (Fig. 11a-c). This zone of
relatively gentle topography coincides with our observations of nearshore/coastal deposits found at approximately 100 m above
sea level. In contrast, the upper half of the runout zone (between 800 and 1300 m along our transect) is characterized by a
sequence of 5 to 7 step-bench segments (Fig. 11c). Steep cliffs of exposed bedrock are defined by east-dipping resistant
graywacke beds that manifest as continuous ledges across the landscape (Fig. 1). The intervening low-gradient (<20°) benches
tend to be broad and approximate bedding planes with a carapace of locally derived colluvium. These steps composed of cliff-
bench sequences are ubiquitous in the marine sedimentary units across Wrangell Island and they are associated with numerous
long-runout landslides in the TNFLI.

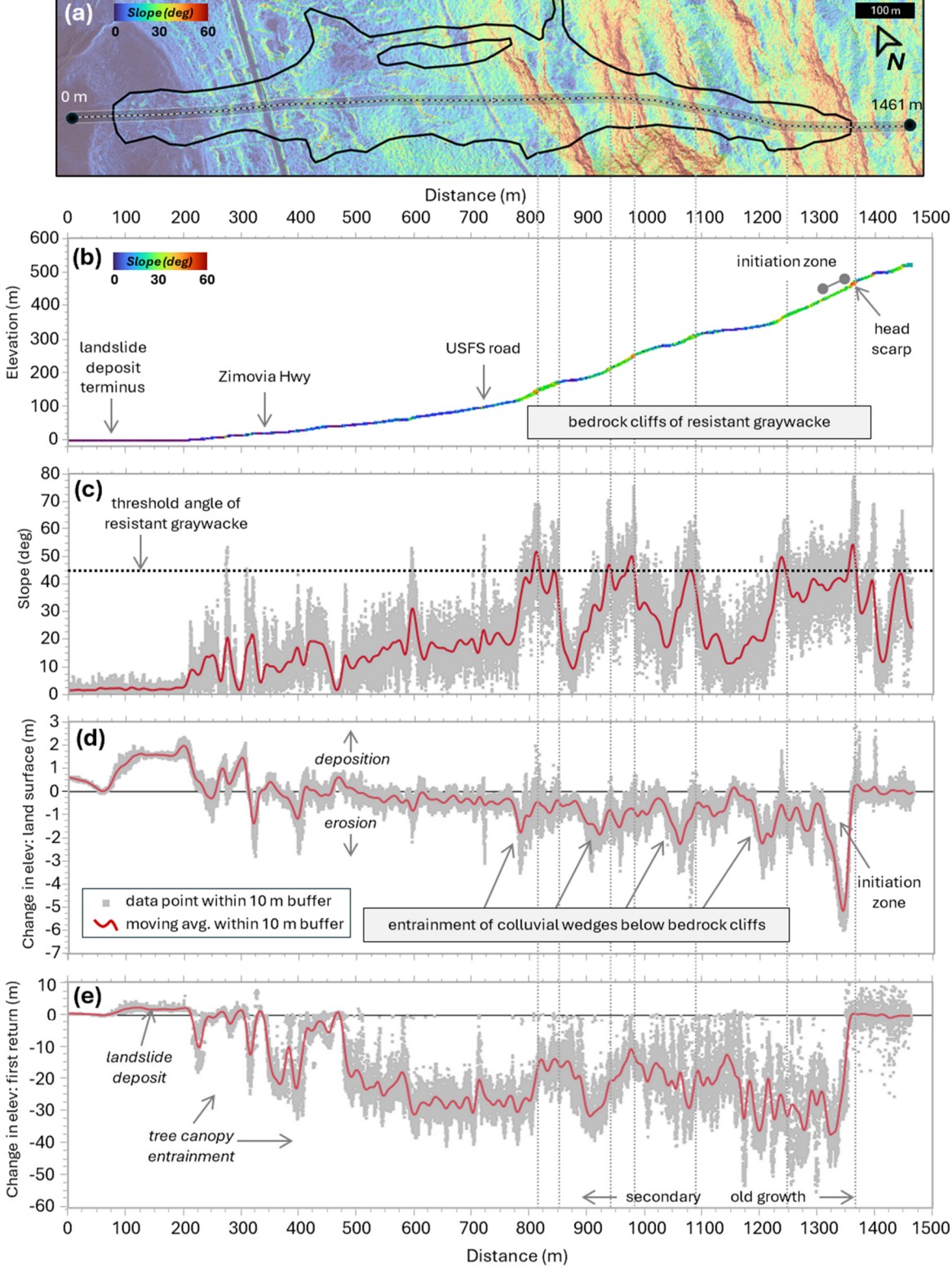

**Figure 11.** Analysis of landslide properties along a 10-m wide longitudinal transect of the MP11.2 landslide. (**a**) Lidar map of slope angle overlain on shaded relief map with transect location and endpoint distances depicted, (**b**) lidar-derived elevation values from the November 2023 acquisition with points coloured by slope angle in degrees, (**c**) lidar-derived slope angle, (**d**) DTM (or land surface) change, and (**e**) DSM (or canopy) change for all points within 10 meters of the transect (grey points) and running average (red line) Secondary and old-growth labels in (e) delineate the boundary between managed and unmanaged forest.

Our profile of DTM (or land surface) change shows that net erosion tends to dominate in the upper half of the landslide while negligible net surface change and deposition characterize the lower half (Fig. 11d). The pattern of erosion in the upper half is strongly correlated with the cliff-bench sequences. Specifically, local erosion maxima of 1 to 2 m (denoted by negative values of surface change) span 25 to 50 m horizontally and occur just downslope of the steep bedrock cliffs where they transition to the low-gradient benches (Fig. 11d). These foci of erosion coincide with field observations of colluvial wedges exposed along the lateral margins of the landslide. Our analysis reveals minimal erosion along the low-gradient benches that are situated below these colluvial wedges, and in the field these benches exhibited patchy entrainment as well as minor local deposition. In the field, we also observed a live blueberry patch growing on a subvertical bedrock face at ~1000 m along the transect (Fig. 9c, 11c). This observation implies negligible erosion, and perhaps projectile behaviour of the landslide runout.

The profile of DSM (or canopy) change indicates removal of trees taller than 50 m in the upper 200 m of the initiation and runout zones, whereas trees less than 40 m in height were mobilized from the lower area of the landslide (Fig. 11e). This pattern results from pre-1965 timber harvests along the lower slopes in our study area with the transition to unmanaged forest at 1,100 m along our transect (Fig. 11e). We sampled cookies from four western hemlock trees transported by the landslide and deposited along the slide margins at approximately 900 m along our transect. The violent nature of the landslide snapped the tree trunks, and we estimated that the lower 3 to 5 m of each trunk was missing. To account for the missing record, we added 20 years to the age of each tree. The four trees ranged from $292 \pm 10$ to $322 \pm 10$ years old, indicating that they originated from the old growth towards the top of the landslide. We also noted that reaction wood (which can be indicative of slope movement) was present in all tree samples (Stoffel et al., 2024).

We plotted average surface or DTM change against local slope for 10-m intervals along the transect to assess topographic controls on debris flow entrainment and deposition (Fig. 12). Net erosion dominates when local slope exceeds 15° and the average value of net erosion increases with slope from 15° to 45°. Notably, points defining this trend occur at a wide range of locations along the transect, reflecting the profound influence of local slope on debris entrainment. That said, locations along the middle section of the landslide, which are denoted by filled green circles (Fig. 12), tend to have lower values of net erosion compared to upslope locations, which may result from variations in debris availability and saturation or changing inertial forces that control entrainment. For slopes between 41° and 44°, we observed several values of high net erosion (>2 m) that deviate from the local slope-erosion trend. These values (denoted by dark red filled circles and a dashed ellipse in Fig. 12) occur at the uppermost extent of the landslide and are associated with the initiation zone and thus reflect mechanical processes that differ

from downslope areas that experienced entrainment. For slope angles less than 15° we observe a trend of increasing deposition
with decreasing slope and a clustering of 0.7 to 1.7 m of deposition at 2° that defines the landslide toe. These trends define the
slope-dependent transition between erosion and deposition for runout models, as well as provide constraints on entrainment

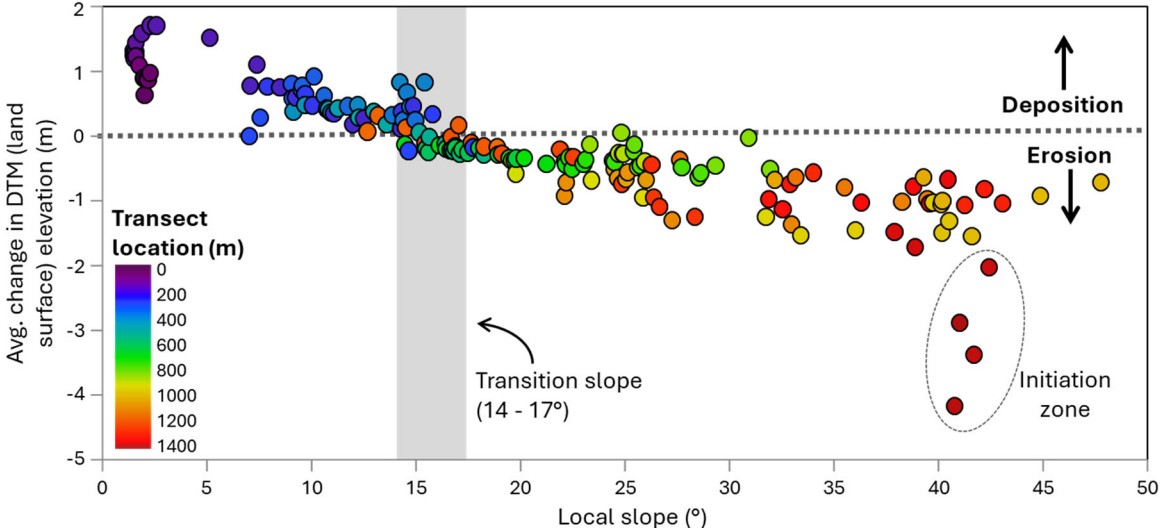

potential.
**Figure 12**. Variation in local net erosion and deposition with slope angle. Values are averaged for 10-m bins along the transect
in Fig. 11a. Colours reflect distance along the transect and the vertical grey rectangle denotes the transition slope between
erosion and deposition. The dark red points enclosed by a gray ellipse denote the initiation zone.
We performed a mass balance of erosion and deposition along the MP11.2 landslide to identify the downslope transition of
net erosion to deposition and quantify the total volume of erosion and deposition associated with the landslide (Fig. 13).
Specifically, we tallied the total thickness of both erosion and deposition for all points within the landslide boundary using 10-
m wide swaths oriented perpendicular to the longitudinal transect (Fig. 11a) and then separately summed the values within
each swath. The distance between distal points along this transect defines the width of the landslide, which averaged less than
100 m in the upper 500 m of the slide, increased abruptly to greater than 200 m through the middle section, and then decreased
to ~100 m in the lower depositional zone (Fig. 13b). Our mass balance analysis indicates high erosion at the initiation zone
that decreased downslope before increasing rapidly just above the middle section, which coincides with landslide widening
(red line in Fig. 13c). In the lower portions of the wide zone (400 to 500 m along the transect), we observe an abrupt transition
from erosion (red line) to deposition (blue line) with a depositional peak that corresponds to the widest section of the landslide.
In the field, this zone of localized widening corresponded with extensive accumulation of downed trees on the north flank of
the landslide. Substantial deposition is associated with the landslide deposit (located between 75 and 250 m along the transect),
just below a zone of local steepness (250 to 350 m along the transect) that experienced efficient transport and minimal
deposition or erosion.

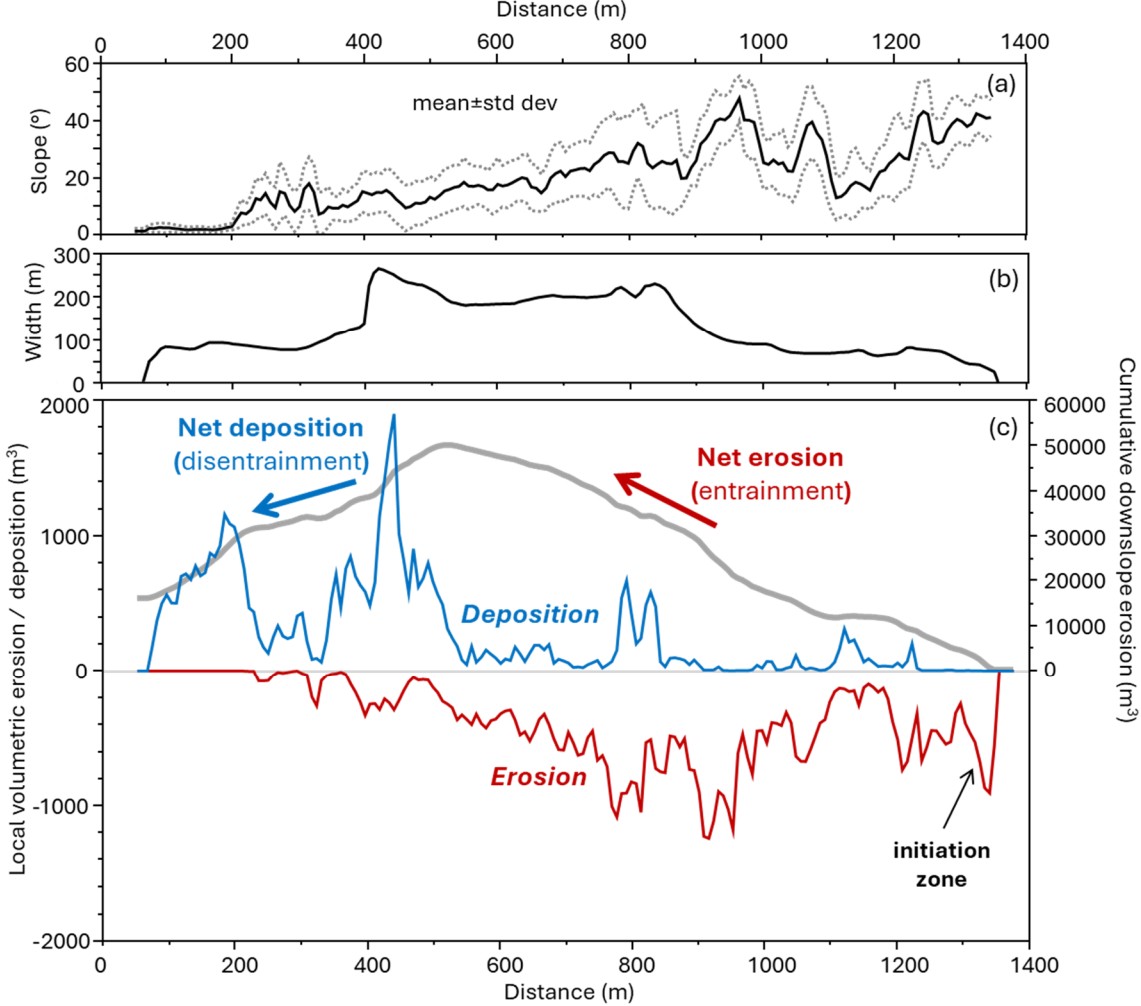

**Figure 13.** Downslope mass balance analysis of the MP11.2 landslide. Profiles of (**a**) mean and standard deviation of slope, (**b**) width, and (**c**) local erosion (red), local deposition (blue), and cumulative erosion minus deposition (gray curve) calculated for DEM cells within the landslide boundaries at 10-m intervals along the transect in Fig. 11a. Note that the northward bifurcation pathway (Fig. 5a) is not included in this analysis.

Lastly, we integrated total erosion and deposition along the landslide path by starting at the head scarp and summing the imbalance in erosion (positive values of erosion) and deposition (negative values of erosion) in the downslope direction (see grey curve in Fig. 13c), finishing at the slide terminus. Cumulative erosion increases monotonically in the downslope direction before peaking at ~550 m along the transect. This implies an average volumetric growth factor of 62 $m^3$ $m^{-1}$ along the erosional portion of the landslide, although local variations associated with changes in slope occur. Downslope of the net erosion peak,

the slide widened and the slope became gentler, and as a result deposition outpaced erosion downslope. At the slide terminus,
the erosion-deposition balance did not approximate zero, however, indicating that net erosion exceeded deposition. In total,
we estimated 65,300 m$^3$ of erosion and 49,400 m$^3$ of deposition for the primary landslide, which implies an imbalance of
>15,000 m$^3$ that may reflect debris loss in the coastal deposition zone as well as detection limits in depositional areas. For the
mass balance of the north-directed bifurcated portion of the landslide (Fig. 5c), we observed 4,000 m$^3$ of erosion and 3,800 m$^3$
of deposition.

## 5 Discussion

### 5.1 Initiation and triggering factors

Our preliminary investigation indicates that the MP11.2 landslide was an anomalously large and thus long-runout event that
initiated in a steep and wide deposit of thick colluvium during a notable but not atypical SE Alaska storm event. A wide range
of factors may have affected the initiation of the landslide and the uncertainty ascribed to our interpretations reflects limitations
in data availability, chiefly local climate observations, to test landslide initiation hypotheses. By contrast, geomorphic and
geologic factors that predisposed the hillslope to landsliding are more straightforward and provide important considerations
for the assessment of landslide hazard and risk in other locations.
Initiation of the MP11.2 landslide likely required a high degree of soil saturation to overcome the shear strength of the
colluvium and promote the observed expansive and fluid-like runout. The rainfall intensity that preceded the landslide was
notable but not extraordinary, as quantified by the 1-yr recurrence interval and 3-hr and 6-hr intensities recorded at Wrangell
airport. Given the 15-kilometer distance between the airport rain gauge and the landslide, and the greater than 400 m elevation
of the initiation zone, the rainfall experienced at MP11.2 is highly uncertain. During our community events, several residents
that drove along the Zimovia Highway on November 20 noted that rainfall south of Wrangell and closer to the landslide area
was more intense than in the town. In addition, several residents reported the presence of a substantial snowpack at mid- and
upper slope locations on the morning of November 20.  At the airport weather station at sea level, air temperatures were cold
(~2°C) on November 19 and warmed rapidly on the morning of November 20, coincident with the arrival of abundant rainfall.
The temporal trend in air temperature at the initiation zone and ridgetop was likely similar although the absolute temperatures
were likely lower owing to the higher elevation. As a result, the rapid warming on November 20 combined with hours of
moderate-intensity rainfall may have generated substantial infiltration and runoff via snowmelt. Planet imagery acquired before
and after November 20 shows changes in snowpack that are consistent with snowmelt contributing to landslide triggering (Fig.
14). Early season high-elevation snow cover shown on November 12 expanded in area, reaching lower elevations and the
landslide headscarp on November 19. By November 28 (and likely sooner), the snow cover was substantially dismissed.
Similarly, in the borough of Haines, Southeast AK, an extreme rain-dominated atmospheric river followed a snow-dominated
atmospheric river in December 2020 generated widespread landslides and the fatal Beach Road landslide (Darrow et al., 2022).
The scale and impact of these recent events suggest that the sequencing and pacing of snow- and rain-dominated storms may
be a critical factor in landslide initiation in SE Alaska. As such, monitoring rain and snow in a wide range of settings is crucial
for advancing our understanding of the hydrologic response that contributes to landsliding.

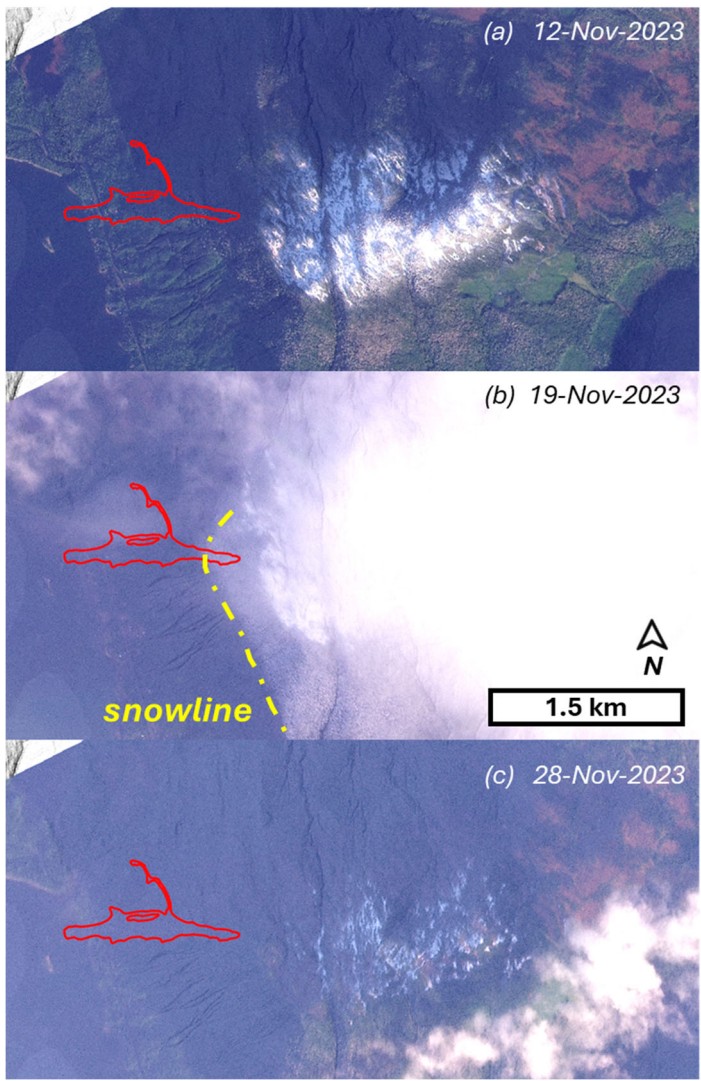

**Figure 14.** Optical satellite imagery showing snow cover before and after the November 20 landslide. **(a)** On November 12,
2023, early season snow resulted in patchy snow cover across the ridgetop wetland that drains to the MP11.2 landslide (red
polygon), **(b)** On November 19, 2023, the snow line (yellow dash-dot line) reached lower elevations, including the headscarp,
and **(c)** On November 28, 2023, the snow cover had thinned substantially. Images © 2023 Planet Labs PBC.

The potential of high wind as a driver of recent landslides across SE Alaska, including the MP11. 2 event, has been surmised
by many residents. Tree turnover (or windthrow) can contribute to the initiation of shallow landslides and debris flows based
on observations from extreme storms (Guthrie et al., 2010; Lin et al., 2025). Such events tend to trigger widespread windthrow,
however, which was not observed on Wrangell Island during the November 20 storm. In the absence of tree turnover, the
potential for trees to transmit dynamic forces into the subsurface due to high winds has not been well-studied although it has
been proposed that vibrations associated with high wind can promote liquefaction (Buma and Johnson, 2015). Alternatively,
windthrow may impact slope stability through the reduction of root reinforcement (Parra et al., 2021). Pioneering research
documenting timber harvest impacts on slope stability was performed in SE Alaska on nearby Prince of Wales Island (Wu et
al., 1979) and those studies demonstrated the substantial contribution of soil shear strength through root reinforcement. More
recent advances highlight how the progressive tensile loading of root systems in shallow soils undergoing shear can be
quantified to assess slope stability in three dimensions, which is critical for capturing how roots reinforce the lateral margins
of potentially unstable slopes (Cohen et al., 2009). These studies demonstrate that as soils get thicker, the relative contribution
of root reinforcement to the total shear strength decreases substantially given that root density decreases exponentially with
depth (Schmidt et al., 2001). The root systems of coniferous forests tend to be concentrated in the upper 1 m (Hales, 2018;
Jackson et al., 1996) and as a result, root reinforcement was likely a minor contributor to the cumulative shear resistance of
the nearly 5-m thick initiation zone of the MP11.2 landslide. Nonetheless, the contribution may not be negligible, and further
analysis of the potential impact of the abating sawfly and budworm infestations on the root systems of western hemlock and
Sitka spruce trees in SE Alaska warrants further investigation. The infestation resulted in moderate-to-severe (11-50%)
mortality of infested trees on Wrangell Island and impacts are common on west-facing slopes and at elevations that coincide
with the initiation zone (U.S. Forest Service, 2025a).
Windy conditions can also contribute to landslide triggering through rapid snowmelt and excess runoff that occurs during
storms with high heat flux, which can be approximated as the product of mean daily temperature and wind speed (Hasebe and
Kumekawa, 1995). Recent analyses of atmospheric rivers have shown that these storms tend to be responsible for extreme
wind, as well as intense rainfall, and approximately half of the top 2% of wind speed events are associated with atmospheric
rivers (Waliser and Guan, 2017). Warm atmospheric rivers in the Sierra Nevada mountains, California, for example, have been
shown to generate a >1 km increase in the snow elevation over several hours, resulting in unanticipated excess discharge,
flooding, and mass movement events (Hatchett, 2018). In 2017, the contribution of extreme wind-driven snowmelt generated
a >35% increase in stream input to the Oroville Reservoir and the excess runoff resulted in overtopping flows and substantial
(>$1 billion) damage to the Oroville Dam as well as thousands of downstream evacuations (Henn et al., 2020). For the MP11.2
landslide, the abrupt rise in temperature and high winds on November 20 combined with the rapid disappearance of higher
elevation snowpack (Fig. 14) imply that wind-driven snowmelt may have contributed to the slope failure by generating excess
runoff and elevated pore pressures in the initiation zone and downstream colluvial wedges. The hydrologic status of the
colluvial materials that were destabilized during the MP11.2 event likely evolve with the combined contributions of antecedent
moisture, rainfall, and snowmelt, although the relative importance of these sources is unclear.
Our field observations of active seepage localized in the SE corner of the MP11.2 head scarp connected to a broad and gentle
ridgetop wetland suggests that the extent and character of terrain above steep slopes may contribute antecedent moisture and
storm runoff that promote landsliding. Our mapping of hydrologic flowpaths along the ridgeline is consistent with these

observations and implies that subtle topographic variability may result in significant changes in the upslope or contributing area of landslide-prone slopes. Similarly, a ponded topographic depression was mapped and monitored upslope of the 2020 Beach Road landslide and narrow channels directly connected that area to the head scarp (Darrow et al., 2022). The abundance of these broad and gentle high-elevation wetlands is highly variable across SE Alaska and likely reflects variations in glacial erosion and bedrock properties (Harris et al., 1974). Combining data from the national wetlands inventory with flow routing analyses provides an opportunity to identify these ridgetop muskeg (or peatland) drainage systems and characterize those with potential to influence hydrologic response on landslide-prone slopes. Because peatlands tend to experience rapid saturation and flashy runoff, they are often sources of storm flow rather than attenuators of high flows (Holden, 2006). As a result, their potential for contributing to landslide triggering demands investigation. Lidar data is a key requirement for characterizing surface hydrology in these environments, and active monitoring of the drainage systems would help determine the magnitude and timescale of hydrologic response and thus the potential contribution to slope instability.

## 5.2 Geologic and geomorphic factors that condition slopes for failure

An additional factor predisposing the hillslopes above Zimovia Highway to landsliding is the accumulation of thick colluvium that constitutes the initiation zone of the MP11.2 landslide as well as downslope material that enabled entrainment and volumetric growth (or bulking) during runout. The thickness of colluvium varies substantially and systematically across the hillslopes. In the field and from our lidar analyses, we observed extensive colluvial wedges draped below resistant graywacke layers of the marine sedimentary unit. The punctuated pattern of downslope entrainment highlights how these colluvial wedges contributed to the volumetric growth and broad area of inundation (Fig. 11d). We interpret these deposits to result from progressive post-glacial rockfall locally derived from the resistant and underlying sedimentary layers.

The combination of east-dipping strata and a west-facing hillslope resulted in the observed pattern of bedrock ledges and thick colluvial wedges that characterize much of the area, and we surmise that a non-negligible difference in bedrock strength may be critical for setting up this geomorphic context. Our Schmidt hammer data highlight the high compressive strength of the graywacke and weak strength of the fine-grained inner beds. At a quarry located just north of the landslide, we documented bedrock structure and observed active slaking of the fine-grained inner beds that may destabilize the overlying resistant beds (supplementary materials). Our kinematic analysis showing favourable conditions for flexural toppling is consistent with our interpretation that progressive failure and retreat of the resistant ledges generate a wake of thick colluvium along the hillslopes (Imaizumi et al., 2015). Importantly, these colluvial wedges will continue to form and thicken with on-going rockfall along the resistant cliffs (Moore et al., 2009) although the pace and frequency of this process is unclear. In nearly all cases, the colluvium is contained within the next downslope bench, which may provide a constraint on the pace of post-glacial bedrock ledge failure and colluvium production. Examination of the TNFLI revealed dozens of other events on Wrangell Island that occurred within a similar geomorphic context. Thus, changes in the bedrock dip and resistance, and slope orientation appear to have a profound effect on the extent and thickness of the colluvial wedges that fuelled the MP11.2 landslide, although further investigation is beyond the scope of this contribution.

## 5.3 Controls on landslide runout and volumetric growth

The large volume and extensive inundation area of the MP11.2 landslide likely originated from a thick and wide initiation zone combined with the entrainment of abundant, saturated colluvium stored on downslope bedrock benches. In this area of SE Alaska, post-glacial isostatic adjustment forms a fringe of uplifted, low-gradient terrain that may provide a key control on landslide runout and deposition. In essence, many landslides on Wrangell and nearby islands appear to terminate upon reaching this low-gradient terrain, when present. Exceptions include particularly large landslides, such as MP11.2, and slides that find and follow confined flowpaths and behave as channelized debris flows. The MP11.2 landslide's depositional slope of 2° is substantially lower than values observed on Prince of Wales and Baranof Islands that vary from 4° to 19° and 6° to 26°, respectively (Booth et al., 2020; Johnson et al., 2000). Given that the mobility value (H/L~0.45) for the MP11.2 slide is not anomalous (Fig. 7a), we interpret its low deposit angle, and thus outsized and tragic impact, to result from highly efficient entrainment and high volumetric growth, which resulted in a large volume and inundation area. Experimental and theoretical investigations of debris flow runout emphasize that pore pressures generated as wet bed sediment is overridden and progressively entrained, can reduce friction and facilitate increases in flow momentum (Iverson et al., 2011; Reid, 2011). These studies emphasize that local slope and volumetric water content are highly sensitive factors that determine the extent of entrainment during landslide runout (Iverson and Ouyang, 2015). Furthermore, because the colluvial stores on the slope were emplaced by rockfall activity and soil transport, they may exist in a contractive state such that deformation and shearing facilitate pore pressure development and volumetric growth.

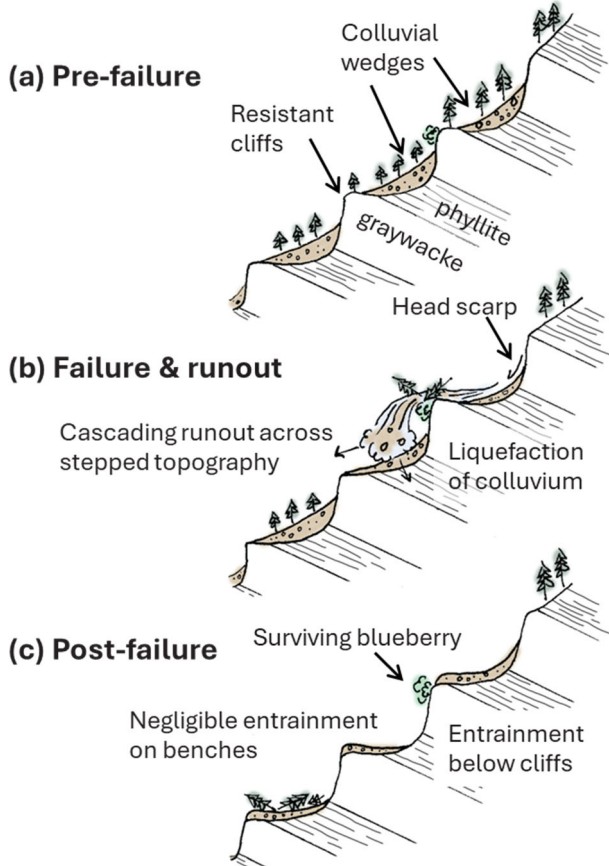

**Figure 15**. Schematic of (**a**) pre-, (**b**) syn-, and (**c**) post-slide hillslope geometry, highlighting the influence of resistant bedrock
and the downslope accumulation of colluvium that becomes mobilized during the landslide event. The live blueberry bush
below a resistant bed reveals cascading, projectile-like behaviour of the slide material.

The volumetric growth factor of 62 $m^3$ $m^{-1}$ is nearly 10x higher than typical values in unglaciated terrain (Reid et al., 2016),
highlighting the importance of unconsolidated sediment thickness on steep slopes for determining landslide volume. From a
mechanistic standpoint, the sequence of subvertical bedrock cliffs along the slide path also suggests that dynamic loading of
stored colluvium from overriding debris may have led to undrained loading and liquefaction (Collins and Reid, 2020). Thus,
in contrast to check dams that are intended to attenuate momentum of flows in mountain channels (Remaître et al., 2008), the
sequence of steps in the MP11.2 runout path may have instead facilitated  runout through a series of loading-induced
liquefaction events (Fig. 15). This behaviour has been noted in other long runout landslides with extensive inundation zones
(Iverson et al., 2015).  Lastly, the 15° slope angle that governs the transition between erosion and deposition along the MP11.2
landslide is steeper than values observed in unglaciated terrain that features valley confinement and thus enables long runout
debris flows (Reid et al., 2016). Acquiring estimates of this transition slope is important for implementing landslide runout
models, such as Laharz and GrfinTools (Brien et al., 2025; Iverson et al., 1998; Reid et al., 2025), and will advance our ability
to predict landslide impacts in the region.

**6 Conceptual framework and research needs for shallow landslide assessment in SE Alaska**

Our analysis highlights key factors that govern the behaviour and hazard potential of shallow landslides in post-glacial
steeplands, such as SE Alaska.
● First, the accumulation of colluvium (or soil) on steep hillslopes serves as a key conditioning process for slope
instability. Previous studies in SE Alaska indicate typical landslide depths of 0.5 to 2.0 m and invoke in-situ
weathering of glacial till, soil creep, and tephra deposition as processes that generate material of sufficient thickness
to initiate shallow landslides (Swanston, 1970). Here, we identify deposition of thick colluvial wedges below resistant
bedrock cliffs as an additional contributor, although the relative importance of these processes remains unclear. More
generally, the timescale of processes that generate colluvium dictates the frequency, magnitude, and spatial pattern
of landsliding in post-glacial landscapes although relevant data are limited.
● Second, characterizing water sources and flow accumulation above landslide-prone hillslopes will facilitate the
identification of terrain with high hazard potential. Many glaciated mountains feature broad, gentle ridgetops that can
store and convey large quantities of surface and near-surface water, particularly during snowmelt and rain-on-snow
events. In British Columbia, this terrain is termed "gentle-over-steep" (Jordan, 2016) and efforts to characterize and
map these particular landforms and quantify drainage patterns using airborne lidar data should be a research priority
in SE Alaska.
● Third, the runout of debris flows and debris avalanches in SE Alaska is seldom facilitated by channels or
topographic confinement. Rather, most landslides traverse poorly-dissected, post-glacial terrain, and the prediction
of debris flow runout in these settings is challenging owing to highly-variable resistance of the surface and flow
materials. In these post-glacial settings, the parameters for empirical models (such as the erosion-deposition transition
angle) have not been constrained and the ability of these models to account for controls on runout is untested.
Physically-based models that account for how large wood and variable grain size dictate flow behaviour also merit
further investigation in conjunction with landslide inventory data and field observations.
● Fourth, because debris flow volume is the primary control on inundation area, quantification of entrainment along
slide paths is essential for runout modelling. The availability of colluvium and its relative saturation can promote
entrainment. Spatial and temporal variations in these two factors likely depend on the pace and pattern of post-glacial
landscape evolution that determines where colluvium accumulates and how hillslope drainage paths are organized.
Thus, landscape evolution models that are developed and tested in postglacial settings should be a research priority.
● Lastly, although atmospheric rivers have been responsible for all the recent fatal landslide events in SE Alaska, the
character and relative magnitude of these ARs have been highly variable. Some have been notable for producing
several hours of intense rainfall while others have been characterized by protracted rain-on-snow. Thus, quantifying
how the sequencing and character of ARs affects landslide susceptibility will be a key component of efforts to build
a landslide warning system (Nash et al., 2024). Currently, the region lacks sufficient weather station stations to capture
strong climatic gradients and climate reanalysis productions (Lader et al., 2020) are limited in scope and resolution.
Most generally, advancing our understanding of how these geomorphic and atmospheric processes contribute to slope
instability across SE Alaska will inform how we assess, plan, mitigate, and manage landslide hazards and minimize impacts
on public safety and infrastructure.
**7 Conclusions**
The 2023 Wrangell Island landslide was among the most devastating and deadly in Southeast Alaska's recent history and
reveals critical insights into shallow landslide processes in post-glacial terrain. Our investigation demonstrates how geological
structure, post-glacial landscape evolution, hydrologic connectivity, and atmospheric forcing combined to produce a high-
impact event with devastating consequences. Although rainfall intensity during the triggering storm was relatively modest, the
landslide magnitude and impact were amplified by several preconditioning factors that are poorly represented with existing
conceptual models and hazard frameworks.
Our key findings include the following:
• Evidence of windthrow contributing to the slope failure is lacking, but rain-on-snow dynamics facilitated by high
wind and warm air temperatures may have delivered critical runoff not captured by typical rainfall intensity
metrics.
• Ridgetop wetlands with subtle drainage divides control hydrologic routing to many landslide-prone slopes,
concentrating surface flowpaths and downslope slope saturation.
• Thick colluvial wedges, perched below resistant bedrock ledges, provided an abundant source zone of readily
mobilized material that fuelled entrainment and long runout.
• The transition between erosion and deposition along the stepped flowpath occurred at 15° regardless of position
along the transect, reflecting the profound influence of local slope angle on sediment entrainment.
• Stepped topography acted to maintain flow momentum, enabling progressive entrainment and promoting long
runout and extensive inundation.
• Sequential lidar and flow modelling are essential tools for identifying landslide initiation susceptibility,
erosion/deposition patterns, and geomorphic preconditioning.
• Large, long runout shallow landslides can occur on anti-dip hillslopes and risk may be greater than previously
recognized.

These findings highlight high key knowledge gaps and can guide future risk mitigation and early warning strategies in steep, post-glacial landscapes. Specifically, advancing landslide prediction in SE Alaska requires expanded lidar coverage, integrated snow and rainfall monitoring, climate modelling, and advances in the modelling of post-glacial landscape evolution, weathering, and colluvium thickness that provide the means for landslide initiation and entrainment.

**Author contributions**

MD wrote the proposal and planned the campaign; MD, JR, AP, and AJ performed the fieldwork; MD collected and analysed the soil and tree samples; JR performed the topographic, climate, and inventory analyses; JR wrote the manuscript draft; MD, JR, AP, and AJ reviewed and edited the manuscript; AP and MD contributed figures and analyses.

**Acknowledgments**

The authors thank the National Science Foundation (RAPID EAR Award 2421234 to University of Alaska-Fairbanks) for supporting this work, Wrangell Cooperative Association for partnership and knowledge sharing, City of Wrangell staff for resources, discussions, and maps, S. and G. Helgesen for access, A. Park and A. Edwards for fieldwork contributions, A. O'Brien and T. Belback for sawyer services, Nolan Center staff for hosting multiple community events, M. Sanders, M. Reid, D. Staley, K. Barnhart, T. Eckhoff, K. Prussian, J. Foss, S. McKay, J. Montigny, and T. Wetor for insightful conversations, and M. Reid, B. Burns, and J. Mey for insightful and helpful review comments.

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
