# Peer review of "Bedrock ledges, colluvial wedges, and ridgetop wetlands"

_EGUsphere, 2025_

## Referee Comment (RC2)

Review for Roering et al. "Bedrock ledges, colluvial wedges, and ridgetop water towers:

Characterizing geomorphic and atmospheric controls on the 2023

Wrangell landslide to inform landslide assessment in Southeast

Alaska, USA"

This study presents an example of a catastrophic shallow landslide in post-glacial terrain in southeastern Alaska. The authors provide a detailed description of the event and adopt a holistic approach to investigate the causes of its initiation and runout behavior. The study is motivated by the frequent occurrence of such events in Alaska and by the existing knowledge gap regarding the triggering mechanisms of shallow landslides in post-glacial landscapes. Ultimately, the work contributes valuable insights for improving landslide risk assessment.

The findings suggest that a combination of several factors contributed to the unique characteristics of this event—namely its unusually high H/L and W/L ratios, large affected area, and high entrainment rate. The most significant factor appears to be the geomorphic setting, where a flat to gently inclined wetland overlies a steep, poorly dissected hillslope. In addition, the step-bench geometry of the slope, resulting from contrasting bedrock strengths, likely facilitated the accumulation of substantial colluvial material that was later remobilized during the landslide. Heavy rainfall, rain-on-snow events, and temperature-induced snowmelt led to oversaturation of the soil layer, serving as the immediate trigger. The potential influence of windthrow and wood pests on root reinforcement is briefly discussed; however, due to limited data, no definitive conclusions can be drawn.

The manuscript is well written and presents a clear, logical progression of ideas from start to finish. I have only a few minor comments: the abbreviation *MP* should be defined upon its first appearance, and the label *NF* in Figure 2 should be made consistent with that used in the caption.

Regards, J. Mey

---

## Author Response (AR1)

Review for Roering et al. " Bedrock ledges, colluvial wedges, and ridgetop water towers:Characterizing geomorphic and atmospheric controls on the 2023Wrangell landslide to inform landslide assessment in SoutheastAlaska, USA"This study presents an example of a catastrophic shallow landslide in post-glacial terrain insoutheastern Alaska. The authors provide a detailed description of the event and adopt a holisticapproach to investigate the causes of its initiation and runout behavior. The study is motivated bythe frequent occurrence of such events in Alaska and by the existing knowledge gap regarding thetriggering mechanisms of shallow landslides in post-glacial landscapes. Ultimately, the workcontributes valuable insights for improving landslide risk assessment.The findings suggest that a combination of several factors contributed to the unique characteristicsof this event—namely its unusually high H/L and W/L ratios, large affected area, and highentrainment rate. The most significant factor appears to be the geomorphic setting, where a flat togently inclined wetland overlies a steep, poorly dissected hillslope. In addition, the step-benchgeometry of the slope, resulting from contrasting bedrock strengths, likely facilitated theaccumulation of substantial colluvial material that was later remobilized during the landslide.Heavy rainfall, rain-on-snow events, and temperature-induced snowmelt led to oversaturation of thesoil layer, serving as the immediate trigger. The potential influence of windthrow and wood pests onroot reinforcement is briefly discussed; however, due to limited data, no definitive conclusions canbe drawn.The manuscript is well written and presents a clear, logical progression of ideas from start to finish.I have only a few minor comments: the abbreviation MP should be defined upon its firstappearance, and the label NF in Figure 2 should be made consistent with that used in the caption.Regards,

**We made the changes requested on Figure 2 and in the text regarding MP.**

The paper is very well written and the authors do a great job investigating and describing the details of what happened.

I have a couple of comments that I think should be mentioned in the paper. The wind. You related the wind to mechanical components like tree throw, but I don't think you mention the wind as an effect on the snow melt. See our DOGAMI SP-55 where we discuss the role of winds in snow melting. The combination of high wind and air temperature increase at the same time can significantly contribute to snow melting and melting rate. The wind blows the warm air into the snow which contributes to the snow melt. We talked with Ben Hatchett about this. If you look at your graph, the wind and air temperature seem to

correlate both high in the time right before the landslide initiates. It is always very hard to say what exactly happened, but I think this is worth mentioning. Is there any way to know how much snow was on the ground days before the event? Even if it was neighbors or roads crews guess. This can clearly affect the terrestrial water input (TWI) above the initiation area, but also how much snow was on the benches? The snow on the benches could play a role in the saturation of the colluvium on the benches. Was there snow on the benches which also underwent rapid melt? I bring this up, because of the lack of likelihood of water from the top of the mountain flowing down the anti-dipping beds and benches and thus not a likely source of saturation of the bench colluvium, which leaves rain from antecedent moisture, rain from this event, and snowmelt all three needing to be directly onto the benches.

Again, really nice paper, authors! This will help Alaskans understand and reduce risk to debris flows.

Bill Burns

Oregon Department of Geology and Mineral Industries

**This is excellent feedback and we addressed the comment by adding a sentence in the introduction as well as an entire paragraph in the discussion section that lays out the potential role of wind in advecting head into snowpacks and facilitating snowmelt. We also addressed the potential means by which colluvial materials on the benches can experience saturation. Many thanks for the helpful reminder and input!**

---

## Author Response (AR2)

**Review#1** for Roering et al. " Bedrock ledges, colluvial wedges, and ridgetop water towers: Characterizing geomorphic and atmospheric controls on the 2023 Wrangell landslide to inform landslide assessment in Southeast Alaska, USA "This study presents an example of a catastrophic shallow landslide in post-glacial terrain in southeastern Alaska. The authors provide a detailed description of the event and adopt a holistic approach to investigate the causes of its initiation and runout behavior. The study is motivated bythe frequent occurrence of such events in Alaska and by the existing knowledge gap regarding the triggering mechanisms of shallow landslides in post-glacial landscapes. Ultimately, the work contributes valuable insights for improving landslide risk assessment. The findings suggest that a combination of several factors contributed to the unique characteristics of this event—namely its unusually high H/L and W/L ratios, large affected area, and high entrainment rate. The most significant factor appears to be the geomorphic setting, where a flat to gently inclined wetland overlies a steep, poorly dissected hillslope. In addition, the step-bench geometry of the slope, resulting from contrasting bedrock strengths, likely facilitated the accumulation of substantial colluvial material that was later remobilized during the landslide. Heavy rainfall, rain-on-snow events, and temperature-induced snowmelt led to oversaturation of the soil layer, serving as the immediate trigger. The potential influence of windthrow and wood pests on root reinforcement is briefly discussed; however, due to limited data, no definitive conclusions can be drawn. The manuscript is well written and presents a clear, logical progression of ideas from start to finish.I have only a few minor comments: the abbreviation MP should be defined upon its first appearance, and the label NF in Figure 2 should be made consistent with that used in the caption. Regards,

Many thanks for the kind words. We made the changes requested on Figure 2 and in the text regarding MP.

**Reviewer #2:** The paper is very well written and the authors do a great job investigating and describing the details of what happened.

I have a couple of comments that I think should be mentioned in the paper. The wind. You related the wind to mechanical components like tree throw, but I don't think you mention the wind as an effect on the snow melt. See our DOGAMI SP-55 where we discuss the role of winds in snow melting. The combination of high wind and air temperature increase at the same time can significantly contribute to snow melting and melting rate. The wind blows the warm air into the snow which contributes to the snow melt. We talked with Ben Hatchett about this. If you look at your graph, the wind and air temperature seem to correlate both high in the time right before the landslide initiates. It is always very hard to say what exactly happened, but I think this is worth mentioning. Is there any way to know how much snow was on the ground days before the event? Even if it was neighbors or roads crews guess. This can clearly affect the terrestrial water input (TWI) above the initiation area, but also how much snow was on the benches? The snow on the benches could play a role in the saturation of the colluvium on the benches. Was there snow on the benches which also underwent rapid melt? I bring this up, because of the lack of likelihood of water from the top of the mountain flowing down the anti-dipping beds and benches and thus not a likely source of saturation of the bench colluvium, which leaves rain from antecedent moisture, rain from this event, and snowmelt all three needing to be directly onto the benches.
Again, really nice paper, authors! This will help Alaskans understand and reduce risk to debris flows.
Bill Burns
Oregon Department of Geology and Mineral Industries

**This is excellent feedback and we addressed the comment by adding a sentence in the introduction as well as an entire paragraph in the discussion section that lays out the potential role of wind in advecting head into snowpacks and facilitating snowmelt. We also addressed the potential means by which colluvial materials on the benches can experience saturation. Many thanks for the helpful reminder and input!**

| | |
|---|---|
| **Associate Editor** I also observed on line 193 the need for a superscript for the cubic meters. Beside that please also consider reviewing the references according to https://www.natural-hazards-and-earth-system-sciences.net/submission.html#references; i saw many pages missing. | **Thanks for the input and third review. Much appreciated. The suggested edits have been implemented.** |

| | |
|---|---|
| **Review #3:** Review of "Bedrock ledges, colluvial wedges, and ridgetop water towers: Characterizing geomorphic and atmospheric controls on the 2023 Wrangell landslide to inform landslide assessment in Southeast Alaska, USA" by Roering et al. This manuscript presents a case study of a recent deadly landslide in Wrangell, Alaska. It describes the geologic materials, geomorphic and hydrologic settings, storm triggering conditions as well as the resulting landslide runout. The authors use modern and valid investigative tools, including repeat lidar, surface hydrologic modeling, detailed mapping, and insightful storm evaluation to investigate controls on landslide initiation and runout. The manuscript also describes noteworthy conditions that facilitated runout, such as liquefaction of colluvial deposits in on a series of downslope bedrock ledges with resulting debris-flow growth – a novel observation that may be relevant to other mobile landslides. Post-glacial hillslopes in Southeastern Alaska have experienced a number of deadly landslides in recent years; thus, causes and conditions leading to this deadly landsliding are of paramount importance. Based on their analyses and interpretations, the authors offer suggestions to aid hazard reduction in these post-glacial environments, such as assessing hillslope colluvial deposits, wind effects from storms, and topographic surface-water controls. The manuscript is well structured, clearly written, and easy to understand. | **Thanks for the helpful summary and encouraging words.** |
| Further clarification of several topics and some editorial modifications would help highlight the key findings in this manuscript. | **Thanks.** |
| Title. Overall, the title is informative. Here are some suggestions for potential improvement: The term "ridgetop water towers" is included in the title, but not defined well in the manuscript. A water tower might be interpreted as a human-made structure, | **The 'water towers' to 'wetlands' is a good suggestion and we've made that** |

| | |
|---|---|
| which is not present at the site. Suggest modifying the title to use "ridgetop wetlands" instead, similar to most other instances in the manuscript or define "water towers" early in the manuscript. Also, the phrase "to inform landside assessment" in the title provides little additional information and could be deleted for a shorter title. | change. The phrase "to inform…" helps differentiate this work from a standard case study and so we kept it. |
| Scientific interpretations. Several items merit clarification in the manuscript. | Thanks. |
| The name of the landslide, MP 11.2, should be mentioned somewhere in the Introduction section and the initials MP defined, before fig. 1. Also specifying its location in the world would be helpful for an international audience. | We define MP11.2 in the introduction and add SE Alaska to figure 1 caption. |
| It would aid understanding to define early in the manuscript both "long runout" and "highly mobile" and then maintain consistency throughout. The landslide is frequently characterized as "long runout" yet not highly mobile. The landslide did runout further than many nearby slides, however It has a larger volume, and therefore should be expected to runout further. Based on the H/L values presented, the landslide was not highly mobile compared to other nearby slides (fig. 7). However, in the Conclusions (line 605) it is inferred that this was a "high mobility" landslide. In addition, it would help orient an international audience to discuss how this landslide's H/L value compares to those for other unconfined debris flows (see Corominas, 1996 for example). | Good point, we've opted to use long runout and not characterize it as highly mobile, which would be consistent with our analysis. Thanks for the heads up on this key point as this resulted in a handful of wording changes in the text. |
| Although the triggering of this landslide occurred during an atmospheric river (AR) metrological event, many AR's on the west coast of North America do not trigger landslides. Is it possible to discuss how this AR differs from other ARs that did not trigger landslides? | The context of this AR is complex and we currently lack the details necessary to characterize the AR properties beyond what's described in the text. A recent paper by Nash and upcoming work by our group will dig into this point but it's beyond the scope of this manuscript. Rather, we will state that a small fraction of ARs trigger slides |

| | |
|---|---|
| | **but most slides are AR-triggered.** |
| The effect of the series of ledges on runout is noteworthy. All things being equal, a stepped topography should act to slow down the slide and reduce runout, not "maintain flow momentum," as mentioned in line 602. Instead (as noted elsewhere in the manuscript), liquefaction of colluvium on each ledge helped to maintain slide momentum, resulting in H/L mobility typical for other nearby landslides. Without this liquefaction, there likely would have been less mobility. This difference should be clarified. | **Good point and we've modified this portion of the text to clarify. Thanks.** |
| Another potential effect from high winds, not mentioned in the manuscript, is tree root vibration, even without actual tree throw (see Swanston, 1974 and Buma and Johnson, 2015 for these effects in SE Alaska). Such vibrations could make soils more likely to liquefy and mobilize into a debris flow. | **That's interesting and we've added it to the text in the discussion section.** |
| Growth and entrainment are an important component of the observed landslide mobility. Entrainment rates typically have a time component. A volume per length growth factor can be described as a yield rate (Hungr et al. 2005), a spatial rate that does not involve time. | **Thanks for the clarification and we've made this distinction clear in the text.** |
| The manuscript analyzes flow directions from ridgetop wetlands. Did water from these wetlands reach the landside area during the triggering storm event or did they provide antecedent moisture to the landslide area? | **Our suggestion is that the flow was potentially a player in both antecedent moisture and storm delivery of moisture to the initiation zone. We've clarified the discussion section text.** |
| Figures. In general, the figures are well crafted and quite informative. Here are some minor suggestions to improve clarity: | **Thanks.** |
| Fig. 1 Add a location map inset showing SE Alaska with landslide location. Suggest adding "SE Alaska, USA" after "Wrangell landslide" in caption. | **Done.** |
| Fig. 2 Add sources of geology and lidar to caption. | **Done.** |
| Fig. 3 Explain the importance of panels (a) and (b) in the text. Add phrase "that triggered the Wrangell landslide" after "event" in the caption. | **Done.** |
| Fig. 5 Add more values to color scales in panels (a) and (b), instead of just min and max. Add lidar resolution to caption. | **We've added the pixel spacing to the** |

| | |
|---|---|
| | **caption but prefer the min/max color bar styling.** |
| Fig. 6 Add star for landslide to panel (b). | **Because the slide is placed within the box plot on panel b, we've opted not to include a start but rather point to it's location on that plot.** |
| Fig. 7 Define "landslide aspect ratio" somewhere. | **Done. We've added definition in the caption.** |
| Fig. 8 Some lidar images (here and elsewhere) appear to have inverted topography (from illumination angle?) leading to upslope curved benches. Suggest modifying images or explaining bench appearance. | **These slopeshade images can sometimes take time to decipher. We see tremendous value in these images because the traditional shaded relief are heavily biased.** |
| Fig. 9 Circle live blueberry bushes as noted several times in manuscript. | **Done.** |
| Fig. 10 Add more values to color scale in panel (b). Add lidar resolution to caption. Note whether size and color of flow arrows denote flux amounts or just flow directions. Also note tendrils mentioned in text (line 341). | **As noted previously, we prefer to keep the min/max color bar labeling scheme. We also clarified that the arrows only relate to direction, rather than amount or magnitude. We also used the caption to clarify the channel vs. un-channel flow lines** |
| Fig. 11 Clarity the extent of "secondary" vs. "old growth" in panel (e), as currently the division between the two is vague. | **Clarified in the caption.** |
| Fig. 12 Ellipse does not appear dashed – different from note in caption. | **The dashes were in fact quite small, so** |

| | |
|---|---|
| | **we clarified the caption.** |
| Fig. 13 Define what is used to compute standard deviation in panel (a), as currently a single transect is implied. Consider modifying the x-axis so that initiation starts at zero, rather than some value between 1200 and 1400 m. Are "all points" in the caption referring to all DEM raster cells or different points? Clarify. | **We clarified the caption for std deviation. We opted not to change the x-axis as we prefer the bottom up reference. All points refer to DEM raster cells, which is also clarified in the caption.** |
| Editorial suggestions and technical clarifications. | **See below.** |
| Line 27 Suggest adding "identifying the distribution of colluvium" to the list of advances needed in Abstract. | **Done.** |
| 34-36 Shallow landslides do not always occur in topographic hollows – they can occur in thin colluvium and also in large topographic amphitheaters. Suggest broadening scope of shallow landslide initiation areas. | **The text has been clarified to be more general.** |
| 103 In the Geology section, it would be useful to mention whether similar geology underlies other recent SE Alaskan landslides. | **We haven't performed a proper analysis and will save this for a future contribution.** |
| 111 Clarify meaning of bedrock cliffs with favorable dip direction, i.e. dip into or out of slope to create cliffs? | **Good catch. Clarified in the text.** |
| 183 Is mid-to-high elevation where the landslide initiated? Clarify. | **Yes, clarified in the text.** |
| 192 Suggest change "noted under" to "noted about". | **Done.** |
| 195 How much is "small but non-negligible"? | **Done.** |
| 209 Add "end of deposit" as that location was used to determine L. | **Done.** |
| 218 Show where the two samples were collected. Fig. 8 shows more than two sediment sample locations. | **The "two" was mistakenly included. Deleted.** |
| 224 Describe type of kinematic analysis. | **Done.** |
| 231 Is high-resolution imagery lidar or photographic? | **Clarified as optical imagery.** |
| 248 Does the initiation elevation refer to the top of the headscarp or the midpoint of the initiation mass? | **Clarified as headscarp** |
| 255 Add Corominas (1996) reference. | **Done.** |

| | |
|---|---|
| 305 Clarify what is "consistent," the deposit existence or the elevation. | **Elevation. Clarified in text.** |
| 315 Add reference for landslides observed in the Tongass National Forest. | **Done.** |
| 324 Add reference for "elevated level of risk in the Sitka area." | **Done.** |
| 356 Quantify "small fraction." | **Done. <5% is the small fraction.** |
| 362 Describe the nature of the coastal deposits. | **They are described earlier in the text.** |
| 393 Add reference for reaction wood. | **Done.** |
| 395 Suggest change "mechanical" to "topographic". | **Done.** |
| 435 Clarify further how 62 m3/m was computed. Entrainment rates usually involve time, this value is similar to yield rate of Hungr et al. (2005) or growth factor of Reid et al. (2016). | **We changed the language to volumetric growth factor and no longer refer to "rate". The description of the calculation has been clarified.** |
| 440 Describe how change in bulk density would account for imbalance. Typically, bulk density decreases in slide material which would increase volume of deposit. | **Good point. Deleted.** |
| 457 Where was the mid-to-upper slope snowpack relative to landslide initiation area? | **New figure with planetary imagery shows the distribution of snowpack relative to the slide, including snow in the initiation zone on the day prior to the landslide.** |
| 461 Did snowmelt run off or infiltrate? | **Probably both. Hard to know. Added "infiltration"** |
| 470 What is location of Beach Road landslide – nearby? | **Clarified. In Haines.** |
| 522 Colluvium may have also been partially saturated – difficult to fully saturate hillslope materials. | **Good point. We clarified by referring to "Positive pore pressures" rather than saturation.** |
| 524 What about termination in ocean, not just low-gradient terrain? | **Although MP11.2 terminates in the ocean, this sentence** |

| | is about most other slides that tend to terminate along the flanks of islands that originate from uplifted shorelines. The text has been clarified. |
|---|---|
| 544 Suggest change "falling" to "overriding." | Done. Good suggestion! |
| 549 Add references for actual models listed – Laharz (Schilling 2014) and Grfin Tools (Reid et al. 2025). | Done. |
| 568 Suggest change "navigate" to "traverse." | Done. |
| 570 Clarify whether resistance is from flow material itself or from ground materials. | Done. Both! |
| 583 Describe what is lacking about weather station observations – spatial and/or temporal resolution? | Done. Focused on strong gradients in climate. |
| 589 Suggest change "impactful" to "destructive." | Done. |

In addition to the revisions spurred by review comments, we've added an additional figure with planet imagery showing the snow cover before and after the landslide event.  We've added this to the discussion section.

---

## Author Response (AR3)

Authors response to comments:

Comment: Figure 2: Please use the proper copyright statement for Bing Maps (see
https://www.microsoft.com/en-us/maps/bing-maps/product/print-rights/)

Response: We changed the Bing image for a Planet labs image and we added the correct
copyright text in the caption.